# Escape from thymic deletion and anti-leukemic effects of T cells specific for hematopoietic cell-restricted antigen

Ji-Min Ju[1,7], Min Ho Jung[1], Giri Nam[1], Woojin Kim[1], Sehwa Oh[2], Hyun Duk Kim[1], Joo Young Kim[3], Jun Chang [3], Sung Hak Lee[4], Gyeong Sin Park[4], Chang-Ki Min[5], Dong-Sup Lee[1], Moon Gyo Kim[6], Kyungho Choi[1] & Eun Young Choi[1,2]

Whether hematopoietic cell-restricted distribution of antigens affects the degree of thymic negative selection has not been investigated in detail. Here, we show that T cells specific for hematopoietic cell-restricted antigens (HRA) are not completely deleted in the thymus, using the mouse minor histocompatibility antigen H60, the expression of which is restricted to hematopoietic cells. As a result, low avidity T cells escape from thymic deletion. This incomplete thymic deletion occurs to the T cells developing de novo in the thymus of H60-positive recipients in H60-mismatched bone marrow transplantation (BMT). H60-specific thymic deletion escapee CD8[+] T cells exhibit effector differentiation potentials in the periphery and contribute to graft-versus-leukemia effects in the recipients of H60-mismatched BMT, regressing H60[+] hematological tumors. These results provide information essential for understanding thymic negative selection and developing a strategy to treat hematological tumors.

[1] Department of Biomedical Sciences, Seoul National University College of Medicine, Seoul 03080, Korea. [2] Interdisciplinary Graduate Program in Genetic Engineering, Seoul National University, Seoul 08826, Korea. [3] Division of Life and Pharmaceutical Sciences, Ewha Womans University, Seoul 03760, Korea. [4] Department of Hospital Pathology, Seoul St. Mary's hospital, The Catholic University of Korea, Seoul 06591, Korea. [5] Department of Internal medicine, Seoul St. Mary's hospital, The Catholic University of Korea, Seoul 06591, Korea. [6] Department of Biological Sciences, Inha University, Incheon 22212, Korea. [7] Present address: Division of Convergence Technology, National Cancer Center, Ilsan, Gyeonggi-do 10408, Korea. Correspondence and requests for materials should be addressed to E.Y.C. (email: eycii@snu.ac.kr)

Thymic negative selection is an important mechanism for the establishment of immune tolerance[1,2]. T cells with specificity for ubiquitous self-antigens are deleted in the thymus to prevent T-cell-mediated autoimmunity[3,4]. In terms of T cells specific for tissue-restricted antigens (TRA) with expression restricted to certain types of cell in the periphery, thymic negative selection is also possible due to promiscuous expression of the TRAs by medullary thymic epithelial cells (mTEC)[5,6]. However, reports have demonstrated that TRA-specific T cells are partially deleted or not deleted at all in the thymus, suggesting that the degree of thymic negative selection differs according to the pattern of antigen distribution[7–10]. Moreover, the fate of T cells that escape thymic deletion varies in the periphery from regulatory T cells to functioning conventional T cells[10,11]. Among these antigens with a cell-type restricted distribution, hematopoietic cell-restricted antigens (HRA) are of particular interest as they are directly presented by thymic dendritic cells (DC). Given the crucial role of DCs in thymic negative selection[12–14], HRA-specific T cells may undergo strict thymic deletion. However, thymic negative selection of HRA-specific T cells has not been addressed in detail, especially using a natural antigen model.

Thymic selection of HRA-specific T cells is also a crucial issue in allogeneic bone marrow transplantation (allo-BMT) for the treatment of hematological malignancies, such as lymphoma and leukemia. In allo-BMT, donor-derived T cells are activated in recognition of allo-antigens displayed in the recipient and eliminate the tumor cells expressing the allo-antigens, generating the graft-versus-leukemia (GVL) effects[15–18]. At the same time, donor T cells can attack the allo-antigen-positive normal tissues in the host, eliciting severe adverse effects and mortality, known as graft-versus-host disease (GVHD)[19,20]. Therefore, allo-antigens expressed exclusively by hematopoietic cells can direct the T cell allo-responses toward the recipient's normal and malignant hematopoietic cells, without eliciting GVHD in the parenchymal tissues, such as the intestine, liver, and skin[17,20,21]. Conventionally, the source of donor T cells responsible for GVL and GVHD was thought to be mature donor T cells contained in the BM inoculum. However, some reports show the mediation of GVHD by donor BM-derived T cells that develop de novo in the thymus of recipients[22]. In animal allo-BMT models, de novo generation of T cells specific for allogeneic TRA and their mediation of GVHD has been demonstrated[23–25]. Thus, it is of value to examine whether HRA-specific T cells that are derived from donor BM and develop in the thymus of the recipient would escape negative selection and mediate GVL without GVHD.

Evaluation of HRA-specific thymic selection requires a natural mouse model HRA and tools to trace the HRA-specific T cells, which are not readily available. Minor histocompatibility antigen (MiHA) H60 is an ideal natural mouse HRA. MiHAs are natural antigens with polymorphism on their peptide fragments presented by MHC I and II, inducing CD8+ and/or CD4+ T cell responses, especially in MHC-matched allogeneic transplantation[26]. H60 is expressed exclusively by hematopoietic cells in the H60-positive strains (i.e., BALB and 129 with $H60^C$ allele), while it is not expressed in C57BL/6 mice for the null mutation at the genetic locus (with $H60^{null}$ allele)[26]. This trait of hematopoietic cell-restricted expression selectively in the H60-positive strains facilitates the establishment of hematopoietic H60-mismatched BMTs using B6 as the BM donor. Moreover, the availability of the H60 congenic mouse strain (B6.C$H60^C$ or Con-H60, hereafter), with an acquired $H60^C$ allele on a B6 background achieved by serial backcrossing, confers B6 T cell development in the H60-positive thymus, excluding any other genetic influences. Because of the null expression of H60 in B6 mice, the H60-specific CD8 T cells are positively selected, with a lack of negative selection in the thymus, and can develop a strong anti-H60 response in the

periphery[27–31]. Thus, comparing the thymic selection of H60-specific T cells in the B6 and Con-H60 thymi would provide clues for understanding the thymic selection of HRA-specific endogenous T cells or donor-derived T cells in the host thymus in allogenic BMT settings. Furthermore, we previously generated H60-cognate TCR transgenic (Tg) mice (named J15) on a B6 background, enabling the fate of H60-specific thymocytes to be traced[32].

In this study, we investigate the development of H60-specific T cells in the thymus where H60 is expressed as a self or allogeneic HRA. Contrary to conventional belief, thymic negative selection of the HRA-specific T cells is incomplete under both endogenous and allogeneic development conditions. T cells with low avidity for hematopoietic H60 that escape the thymic deletion have effector differentiation potential in the periphery. Moreover, these surviving mature T cells contribute to the regression of H60+ hematological tumors in the recipients of H60-mismatched BMT. These results deepen our understanding of the thymic negative selection of TRAs including HRAs and enable the development of a strategy to enhance the treatment of hematological tumors.

## Results

**Partial deletion of J15 thymocytes cognate for self-HRA H60.** To investigate the thymic negative selection of T cells specific for self-hematopoietic antigen H60, we analyzed the thymic development of the progenies of crosses between J15 Tg and Con-H60 mice, J15$^{Tg}$Con-H60 ((J15×Con-H60) F1, hereafter). J15 TCR α and β chains (Vα10.3 and Vβ8.3) originate from a CD8 T cell clone with high avidity for H60[32,33]. Cellularity in the (J15×Con-H60) F1 thymi and the fractional representations of CD4+CD8+ double positive (DP) and CD4−CD8+ single positive (SP) cells were significantly lower, compared to those in the (J15×B6) F1 thymi, which indicated the presence of negative selection of the J15 thymocytes in the former (Supplementary Fig. 1a and Fig. 1a). But the negative selection was not complete, as an average of 20% of the CD8+ SP cells from (J15×Con-H60) F1 thymi showed positive staining for H60-tetramers. This was in contrast to the full deletion of J15 thymocytes in the thymi of F1 progenies obtained from crossing with H60 Tg mice (Act-H60, hereafter) in which H60 is expressed ubiquitously under the control of the actin promoter[34]. In the CD8+ SP cells from thymi of J15$^{Tg}$Act-H60$^{Tg}$ mice ((J15×Act-H60) F1, hereafter), such H60-tetramer-binding (H60-tetramer+) cells were barely detected.

Comparison of splenic profiles between the three different F1 mice demonstrated that the frequencies and numbers of H60-tetramer+ cells in the splenic CD8 T cells from (J15×Con-H60) F1 mice were significantly lower than those from (J15×B6) F1 mice but were significantly higher than those from (J15×Act-H60) F1 mice (Fig. 1b), which reflected the different levels of negative selection of the J15 thymocytes in the respective F1 mice. It was interesting to note that the mean fluorescence intensities (MFIs) of the H60-tetramer staining were significantly lower for the H60-tetramer+ cells in the CD8+ splenocytes and thymocytes from (J15×Con-H60) F1 mice compared with the corresponding H60-tetramer+ cells from (J15×B6) F1 mice (Fig. 1a, b). The lowered tetramer staining intensity in the H60-tetramer+ CD8+ cells from (J15×Con-H60) F1 mice was not an artifact generated by overlap between the binding epitopes of anti-TCR Vβ8.3 antibody and H60-tetramers (Supplementary Fig. 2a). Rather, it could be explained by pairing of the transgenic J15 TCR β chain with endogenous TCR α chains due to incomplete allelic exclusion. Alternatively, downregulation of TCR and CD8 on the surface of J15 T cells by continuous antigen stimulation in the Con-H60 thymus and spleen could permit escape from negative selection

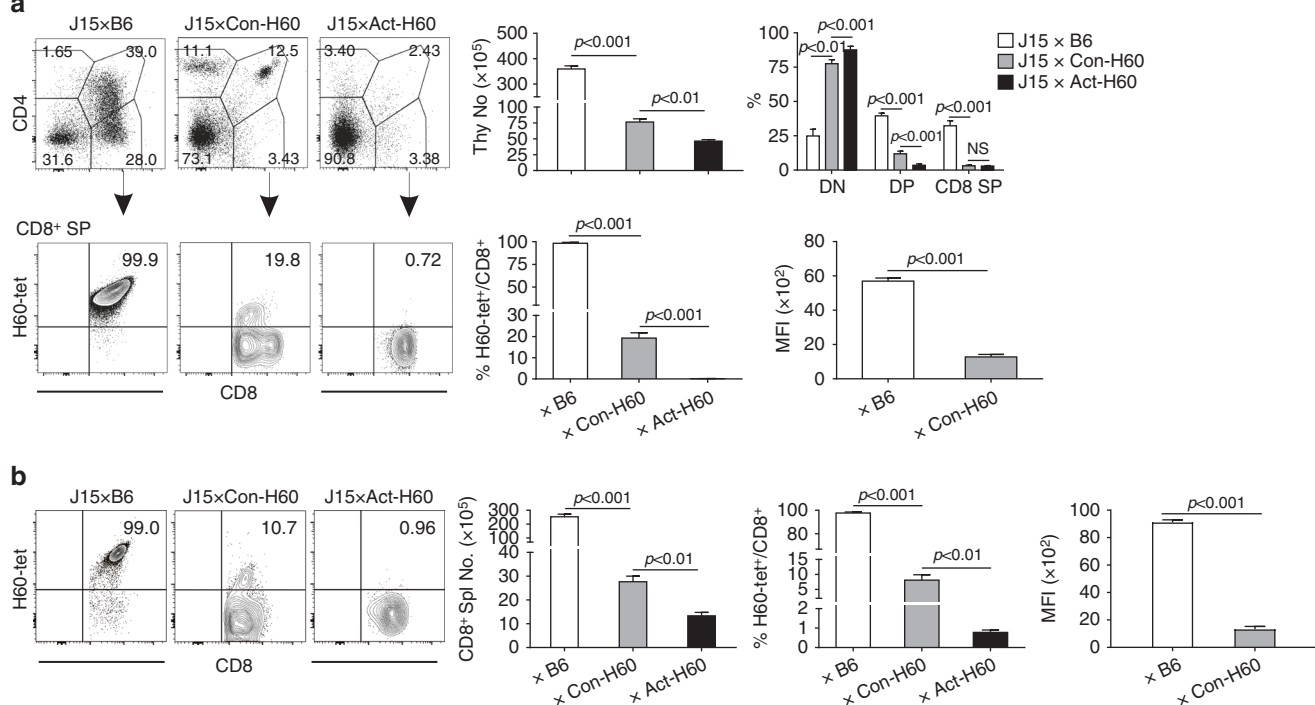

**Fig. 1** Self-hematopoietic antigen H60 derives partial negative selection of J15 thymocytes. **a** Flow cytometric analysis of thymocytes from F1 progenies of crossing J15 mice with B6, Con-H60, and Act-H60 mice. Representative CD4-phycoerythrin.Cychrome 5 (PE.Cy5) by CD8-allophycocyanin.Cychrome 7 (APC.Cy7) FACS profiles are shown with gating on Vβ8.3$^+$ cells. H60-tetramer-PE by CD8-APC.Cy7 FACS profiles are shown after further gating on CD8$^+$ SP cells. The percentages are indicated in the FACS profiles. The number of total thymocytes; percentages of CD4$^-$CD8$^-$DN, CD4$^+$CD8$^+$ DP, and CD8$^+$ SP cells in the total thymocytes; and percentages and MFI values of H60-tetramer$^+$ cells in CD8$^+$ SP cells were plotted. **b** Analysis of splenocytes from the three different J15 F1 mice. The numbers of splenic CD8$^+$ T cells and the frequencies and MFI values of H60-tetramer-binding cells among the splenic CD8 T cells are plotted. Representative FACS data show H60-tetramer-PE staining in splenic CD8 T cells after gating on Vβ8.3$^+$; the percentages of H60-tetramer-binding cells are indicated. All data (**a**, **b**) represent more than three independent experiments (n = 3/group/experiment) and are presented as means ± s.e.m. P-values were generated by Student's t-test

due to low avidity antigen-recognition and maintain the low avidity of the T cells in the periphery[35–37], although these two explanations are not mutually exclusive. These interpretations were supported by the increased frequency of endogenous α chain-harboring cells in the H60-tetramer$^+$ CD8$^+$ T cell populations, and also by the reduced expression of TCR/CD8 and elevated expression of activation markers, CD69/CD44, on tetramer$^+$ CD8$^+$ cells from (J15×Con-H60) F1 thymus and spleen, compared with their (J15×B6) F1 counterparts (Supplementary Fig. 2b, c). Collectively, these results indicated that J15 thymocytes were partially deleted in the thymus where H60 is expressed as a self-HRA, suggesting escape from thymic deletion and development into mature CD8 T cells of thymocytes with low avidity for hematopoietic H60.

**Incomplete thymic deletion against allogeneic HRAs.** Given the identity of H60 as a transplantation antigen, we focused on the negative selection of J15 thymocytes newly developing in the thymus of allogeneic recipients where H60 is expressed as an HRA, performing H60 single antigen-mismatched BMT. For this, lethally irradiated Con-H60 mice were transplanted with BM cells isolated from CD45.1$^+$J15 mice (J15→Con-H60; Fig. 2a). Because irradiation pre-conditioning induces death of hematopoietic cells in BMT recipients, negative selection of J15 thymocytes developing de novo in the thymus of Con-H60 recipient was expected to be severely compromised. However, CD45.1$^+$Vβ8.3$^+$ cell-gated flow cytometric analysis (Supplementary Fig. 1b) showed that incomplete deletion of donor-derived J15 thymocytes developing

in the thymi of Con-H60 recipients occurred. H60-tetramer$^+$ cells were detected in the CD8$^+$ SP thymocytes, although fractional representations and numbers of the CD4$^+$CD8$^+$ DP and CD8$^+$ SP cells were significantly low in the thymi of Con-H60 recipients (Fig. 2b).

When BM cells from Rag-1-deficient (Rag-1$^{-/-}$) CD45.1$^+$J15 mice were transplanted (Rag-1$^{-/-}$ J15→Con-H60), to preclude endogenous TCRα expression by J15 thymocytes developed de novo, incomplete deletion of J15 thymocytes was reproduced (Fig. 2c). However, the frequencies of H60-tetramer$^+$ cells in the CD8$^+$ SP thymocytes were remarkably enhanced (91.3%, on average), due to a lack of T cells expressing endogenous TCRα (Supplementary Fig. 2d). H60-tetramer$^+$ CD8$^+$ SP thymocytes from Con-H60 recipients of Rag-1$^{-/-}$ J15 BM also showed less intense tetramer staining with reduced surface expressions of TCR and CD8 compared to those from the B6 counterparts (Supplementary Fig. 2e), which supports the idea that TCR downregulation is a mechanism for escape of J15 T cells from thymic deletion.

Next, using (Act-H60→B6) BM chimeras as recipients of CD45.1$^+$J15 BMT, we evaluated whether the incomplete deletion of HRA-cognate thymocytes could be reproduced in other models (Fig. 2d). In the recipient mice of this model, H60 is expressed only by the hematopoietic cells derived from Act-H60 BM. J15 thymocytes were incompletely deleted in the J15→(Act-H60→B6) BMT, as well as in the J15→(Con-H60→B6) control BMT (Fig. 2e). Moreover, OT-1 thymocytes were also incompletely deleted in OT-1→(Ova Tg→B6) BMT, in which ovalbumin expression is restricted to the recipient's hematopoietic cells (Supplementary

Fig. 3). Therefore, incomplete thymic negative selection is applicable to other HRA-specific CD8 T cells under BMT conditions.

**Different stages of deletion in Act-H60 and Con-H60 thymi.** J15 thymocytes are negatively selected at the CD4⁻CD8⁻double negative (DN) developmental stage in Act-H60 mice[38]. To understand the reason underlying the different levels of deletion

of J15 thymocytes in the Con-H60 and Act-H60 recipients, we analyzed the CD4⁻CD8⁻DN thymic profiles of the two H60⁺ recipients based on the CD25/CD44 phenotypes (Supplementary Fig. 1c). CD45.1⁺lineage⁻ (Lin⁻; B220⁻NK1.1⁻CD11c⁻CD11b⁻) CD4⁻CD8⁻DN cells of $Rag^{+/+}$ or $Rag^{-/-}$ J15 thymocytes from Con-H60 recipients were found to consist of DN1 (CD25⁻ CD44⁺) through DN4 (CD25⁻CD44⁻) cells, as DN4 cells were detected in the DN thymocytes from the B6 counterparts (Fig. 3a and Supplementary Fig. 4a). However, the DN4 fraction in the

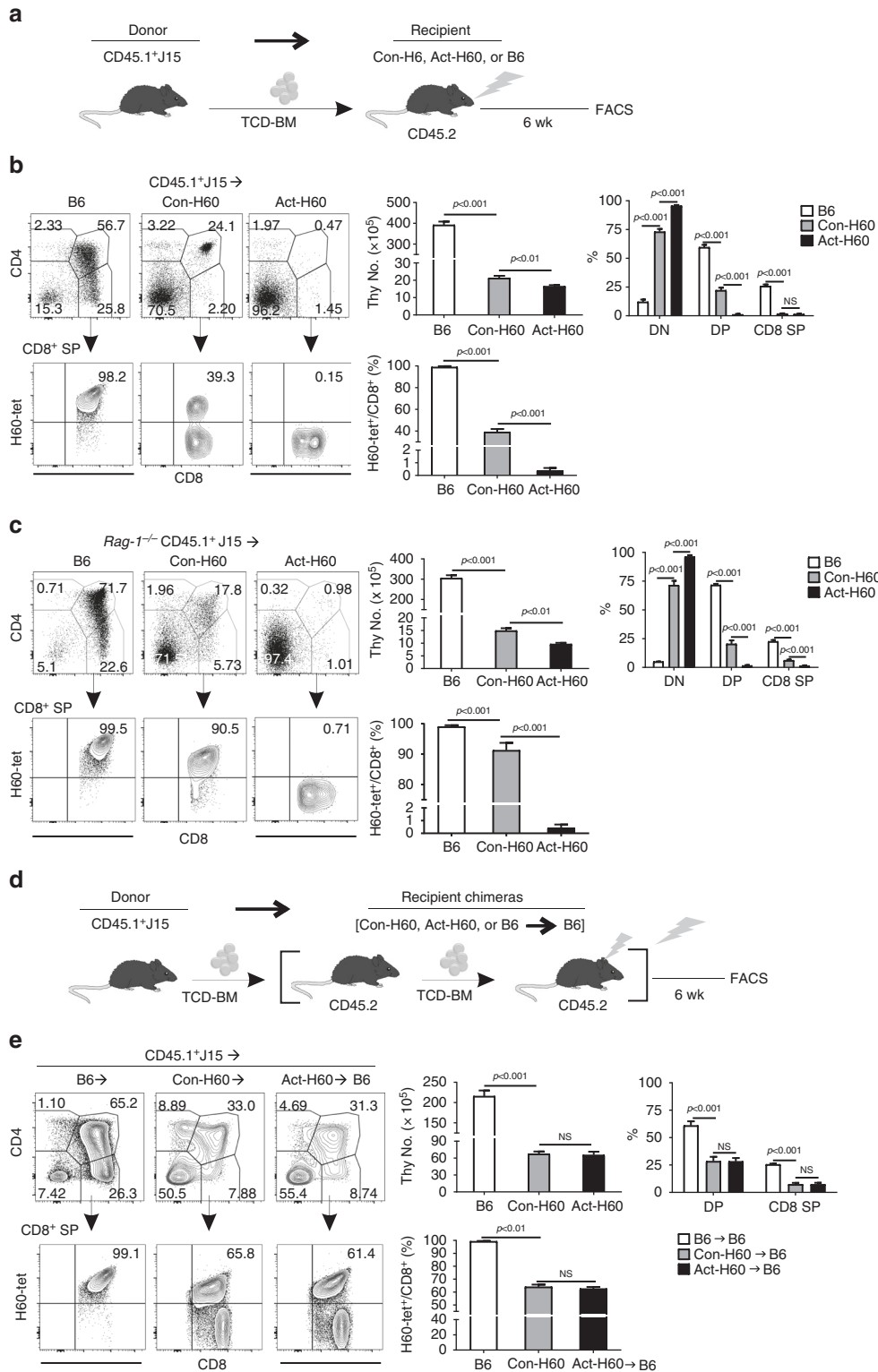

DN thymocytes from Con-H60 recipients was comparatively reduced. On the other hand, DN thymocytes from the Act-H60 recipients lacked post-DN2 stage cells.

When surface levels of CD5 were compared between different stages of thymocytes from the three different recipients, significant upregulation of CD5, indicative of the presence of TCR-signaling events in thymocytes[39], was observed on the DN4 and DP cells from Con-H60 recipients, while it was observed on the DP or DN2 cells from B6 or Act-H60 recipients, respectively (Fig. 3b and Supplementary Fig. 4b). When stained with antibody against active Caspase-3 to detect thymic deletion[40], significant fractions of the DN4 and DP cells from Con-H60 recipients and DN2 cells from Act-H60 recipients were positive (Fig. 3c), indicating that J15 thymocytes were deleted at the CD5 upregulation stages in both negatively selecting thymic environments. The DN4-DP stages of CD5 upregulation with reduced DN4 fraction was reproduced with J15 thymocytes in other BMT experimental models including the J15→(Act-H60→B6) and J15→(Con-H60→B6) BMTs (Fig. 3d, e), and also with the J15 thymocytes developed in the (J15×Con-H60) F1 thymus (Supplementary Fig. 4c, d). Collectively, these results demonstrated that J15 thymocytes underwent negative selection at different developmental stages in the two H60[+] recipients, at the DN4-DP stages in the Con-H60 thymus and the DN2 stage in the Act-H60 thymus, and the incomplete deletion of J15 thymocytes for hematopoietic H60 showed a correlation with CD5 upregulation at their DN4-DP cells.

**Hematopoietic APCs of Con-H60 host delete J15 thymocytes.** The results above suggested that DCs, which are localized mainly in the cortico–medullary junction and the medulla of the thymus, would be responsible for the negative selection of J15 thymocytes at the DN4-DP stages in the thymus of Con-H60 recipients, and the cells localized in the cortex, probably cortical TECs (cTECs), at the DN2 stage in the thymus of the Act-H60 recipient. DCs are able to delete mTEC-selected antigen-specific thymocytes[4,14,41,42]. As H60[+] hematopoietic cells are susceptible to irradiation-induced death in the Con-H60 recipients, donor BM-derived DCs could act as major deleting cells for J15 thymocytes, presenting H60 derived from dying host hematopoietic cells. To test this, we explored whether blocking of antigen presentation by donor-derived DCs would abolish the deletion, performing $\beta 2m^{-/-}$ J15→Con-H60 BMT. However, the incomplete deletion of J15 thymocytes, with CD5 upregulation at the DN4-DP stages, persisted in this $\beta 2m^{-/-}$ J15→Con-H60 BMT (Fig. 4a–e, left). This indicated that donor-derived DCs were dispensable in the deletion of J15 thymocytes in the thymus of Con-H60 recipient, suggesting that the radiation-resistant host hematopoietic APCs and/or stromal APCs ectopically expressing H60 might be major players in this deletion.

CD45.1[−]MHCII[+]CD11c[+] host DCs remained in the thymus after CD45.1[+]J15→Con-H60 BMT, accounting for 0.2–0.3% of thymic cellularity (Supplementary Fig. 5a). To leave only stromal cells as intact APCs in the Con-H60 thymus, $\beta 2m^{-/-}$J15→ ($\beta 2m^{-/-}$→Con-H60) BMT was performed. Thymic profiles of the recipients indicated that positive rather than negative selection occurred to the J15 thymocytes in this $\beta 2m^{-/-}$J15→ ($\beta 2m^{-/-}$→Con-H60) BMT: the fractional representations of thymocytes at different developmental stages and their CD5 upregulation dynamics were comparable to those observed in the $\beta 2m^{-/-}$J15→($\beta 2m^{-/-}$→B6) control BMT (Fig. 4a–e, right). However, full deletion of J15 thymocytes, with CD5 upregulation at the DN2 stage, persisted in a $\beta 2m^{-/-}$J15→($\beta 2m^{-/-}$→Act-H60) control BMT. Thus, H60[+] host hematopoietic cells and cTECs were indispensable for deletion of J15 thymocytes in the thymi of Con-H60 recipients and Act-H60 recipients, respectively. Indeed, H60-transcript was detected only in the thymic DCs, not in the mTECs, of Con-H60 mice by RT–PCR (Supplementary Fig. 5b, c).

Consistently, when antigen presentation was prevented in the cells of recipients by performing J15→$\beta 2m^{-/-}$ Con-H60 or J15→ $\beta 2m^{-/-}$Act-H60 BMT, the incomplete or complete deletion, respectively, of J15 thymocytes was disturbed. Despite the proper localization of donor-derived β2m[+] cells in the thymi of the recipients, J15 thymocytes in the two BMTs exhibited profiles similar to those after J15→($\beta 2m^{-/-}$B6) BMT, including an increased DP population and CD5 upregulation at the DP stage (Fig. 4f–j). These results confirmed that H60[+] host hematopoietic APCs were responsible for the deletion of J15 thymocytes in the Con-H60 recipients, although not the complete deletion.

**Effector potentials of J15 escapee CD8[+] T cells.** Next, we tested whether the J15 escapee CD8 T cells generated de novo in the periphery could differentiate into effector cells. Although the average number of H60-tetramer[+] CD8 T cells in the spleens of Con-H60 recipients was significantly lower than that of B6 recipients, the H60-tetramer[+] splenic CD8 T cells from Con-H60 recipients produced IFN-γ in response to in vitro stimulation with H60-peptides at concentrations above $10^{-10}$ M; this was also the case for those from the B6 recipients (Fig. 5a, b). However, the levels of IFN-γ[+] cells were significantly lower in the H60-tetramer[+] splenic CD8 T cells from Con-H60 recipients than in those from B6 recipients (Fig. 5b). A similar tendency was observed in the H60-tetramer[+] CD8 T cells from (J15×Con-H60) F1 and (J15×B6) F1 mice (Supplementary Fig. 6a, b), although the basal levels of IFN-γ[+] cells were much lower in the endogenously generated J15 escapees in the F1 mice. The J15 escapee CD8 T cells proliferated in response to H60-stimulation in vitro when subjected to mixed leukocyte culture (MLC) with irradiated H60[+] feeder cells, albeit with delayed kinetics and lower level of BrdU incorporation compared with J15 CD8 T cells from B6 recipients

**Fig. 2** Incomplete thymic negative selection of J15 T cells specific for HRA H60. **a** A schematic illustration depicting the experimental design. T-cell-depleted (TCD) BM cells (5×10[6]) from CD45.1[+]J15 mice were transplanted into lethally irradiated (dose: 1000 cGy) B6, Con-H60, and Act-H60 mice. Next, CD45.1[+]Vβ8.3[+] donor BM-derived thymocytes and splenocytes were analyzed by flow cytometry at 6 weeks following BM transplantation. **b** Flow cytometric analysis of thymocytes from the CD45.1[+]J15 BMT recipients. Representative CD4-PE.Cy5 by CD8-APC.Cy7 FACS profiles are shown with gating on CD45.1[+]Vβ8.3[+] cells. H60-tetramer-PE by CD8-APC.Cy7 FACS profiles are shown after further gating on CD8[+] SP cells. The number of total thymocytes; percentages of CD4[−]CD8[−]DN, CD4[+]CD8[+] DP, and CD8[+] SP cells in the total thymocytes; percentages of H60-tetramer[+] cells in CD8[+] SP cells were plotted. **c** Flow cytometric analysis after transplantation of Rag-1[−/−]CD45.1[+]J15 BM cells. Representative FACS data of CD4-PE.Cy5/CD8-APC. Cy7 profiles and H60-tetramer-PE staining in CD8[+] SP cells are shown. Data were processed as described above in **b**. **d** A schematic illustration of the experimental design. BM chimeric recipients were generated by the transplantation of 5×10[6] TCD BM cells from B6, Con-H60, and Act-H60 mice into lethally irradiated B6 mice. Six weeks after this primary BMT, 5×10[6] TCD BM cells from CD45.1[+]J15 mice were transplanted into lethally irradiated (B6→B6), (Con-H60→B6), and (Act-H60→B6) BM chimeras. CD45.1[+]Vβ8.3[+] donor BM-derived thymocytes were analyzed by flow cytometry at 6 weeks post-BM transplantation. **e** Flow cytometric analysis of thymocytes from double BMT recipients. Representative FACS profiles of CD4-PE.Cy5/CD8-APC. Cy7 staining and H60-tetramer-PE staining in CD8[+] SP cells are shown. Bar graphs were processed as described above in **b**. Data from **b**, **c**, **e** represent more than five independent experiments (n = 3/group/experiment) and are presented as means ± s.e.m. P-values were determined by Student's t-test

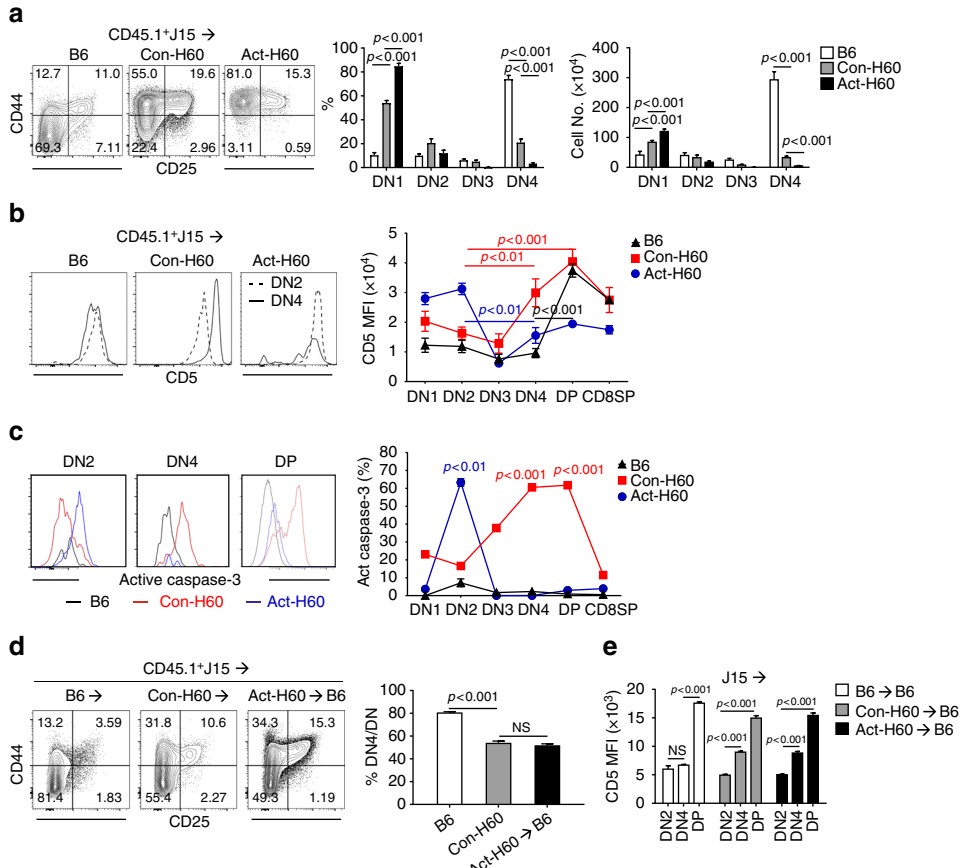

**Fig. 3** Delay in thymic negative selection of J15 T cells in Con-H60 recipients. **a** Representative flow cytometric analysis of CD4⁻CD8⁻DN thymocytes in the recipients of CD45.1⁺J15 BMTs. CD44-PE.Cy7/CD25-allophycocyanin FACS data are shown after gating on CD45.1⁺Lin⁻CD4⁻CD8⁻cells. Representative FACS data values indicate the percentages of each quadrant fraction in the DN cells. These percentages and the corresponding cell numbers are plotted as bar graphs. DN1, DN2, DN3, and DN4 cells indicate the CD44⁺CD25⁻, CD44⁺CD25⁺, CD44⁻CD25⁺, and CD44⁻CD25⁻ quadrants, respectively. **b** CD5 expression profiles in thymocytes at each stage. Representative FACS data shown as single histograms of CD5 expression in DN2 and DN4 thymocytes from three different recipients. CD5-PE MFI values in thymocytes of each stage are plotted. **c** Detection of active Caspase-3 in thymocytes from Con-H60 and Act-H60 recipients. Single histograms represent FACS data after staining with anti-active Caspase-3-PE. Percentages of active Caspase-3⁺ cells among thymocytes at different development stages were plotted. **d** DN cell FACS profiles in (B6→B6), (Con-H60→B6), and (Act-H60→B6) recipients of CD45.1⁺J15 BMT. Representative CD44-PE.Cy7/CD25-allophycocyanin profiles are shown. The frequencies of DN4 cells among DN cells are plotted. **e** MFI values of CD5-PE surface staining on DN2, DN4, and DP cells are plotted. All data (**a**–**e**) are representative of three (**c**) and more than five (**a**, **b**, **d**, **e**) independent experiments (n = 3/group/experiment). Data are presented as means ± s.e.m. P-values were generated by Student's t-test

(Supplementary Fig. 7a). Taken together, these results indicated that T cells of low avidity for hematopoietic H60 escaped thymic negative selection and entered the periphery with the potential to proliferate and differentiate into IFN-γ producing effectors in response to H60 stimulation.

In the periphery of Con-H60 recipients, H60-tetramer⁺ escapee CD8 T cells were the major T cell subset infiltrating the organs and were mixed with CD8αβ and CD8αα cells (Supplementary Fig. 7b; Fig. 5c). The presence of CD8αα cells was pronounced in the non-lymphoid organs including the intestines, which is consistent with a previous report on the presence of non-pathologic CD8αα dimeric thymic deletion escapees in the intestines[43]. The H60-tetramer⁺ escapee CD8 T cells were mostly CD44^hi in all of the tested peripheral organs (Fig. 5d). When peripheral leukocytes were typed for their origins, those from Con-H60 recipients showed enhanced donor chimerism, with a reduced CD45.1⁻ host cell fraction, compared with those from B6 and Act-H60 recipients (Fig. 5e). This result implied that CD45.1⁻ host hematopoietic cells would be eliminated by the cytotoxic function of H60⁻tetramer⁺ escapee CD8 T cells that were present

in activated status in the peripheral organs of Con-H60 recipients. Consistently, when peptide-loaded and CFSE-labeled target cells were intravenously (i.v.) injected, Con-H60 recipients showed more efficient killing of H60-peptide-loaded target cells (CFSE^hi) compared to B6 recipients (Fig. 5f). The Foxp3⁺ cell fractions in splenic and intestinal CD4 and CD8 T cells were not significantly different between Con-H60 and B6 recipients (Supplementary Fig. 7c). Taken together, these results enabled us to conclude that J15 escapee CD8 T cells had effector potential to kill H60⁺ cells in the periphery.

**Contribution of J15 CD8⁺ escapee T cells to GVL effect.** Next, we explored whether the J15 escapee CD8 T cells with effector potential could exert GVL effects against H60⁺ hematological tumors. EL4 cells transduced to express H60 (H60⁺EL4) or the cells transduced with an empty vector (H60⁻EL4) were subcutaneously (s.c.) injected into Con-H60, B6, or Act-H60 recipients of J15 BMT (Fig. 6a). Con-H60 and B6 recipients, which were implanted with H60⁺ EL4 cells, showed long-term survival

without tumor outgrowth, while all the other experimental groups, including the H60+EL4-implanted Act-H60 recipients, died from their tumors within 30 days after tumor injection (Fig. 6b). In a minor population (20–30%) of Con-H60 recipients, occasional mortality with weight loss and diarrhea was detected (Supplementary Fig. 8a, b). As this occurred regardless of whether H60+ EL4 or H60−EL4 tumor cells were injected, the sporadic

incidence of intestine GVHD-like symptoms was ascribed to the high infiltration of H60-tetramer+ escapee CD8αβ T cells specifically into the intestines of Con-H60 recipients, which was consistent with the presence of the target antigen (H60-bearing leukocytes) in the intestine, and significant expression of the gut-homing receptor integrin α4β7[44,45], but not the liver-homing receptor CXCR6[46], by the splenic H60-tetramer+ escapee T cells,

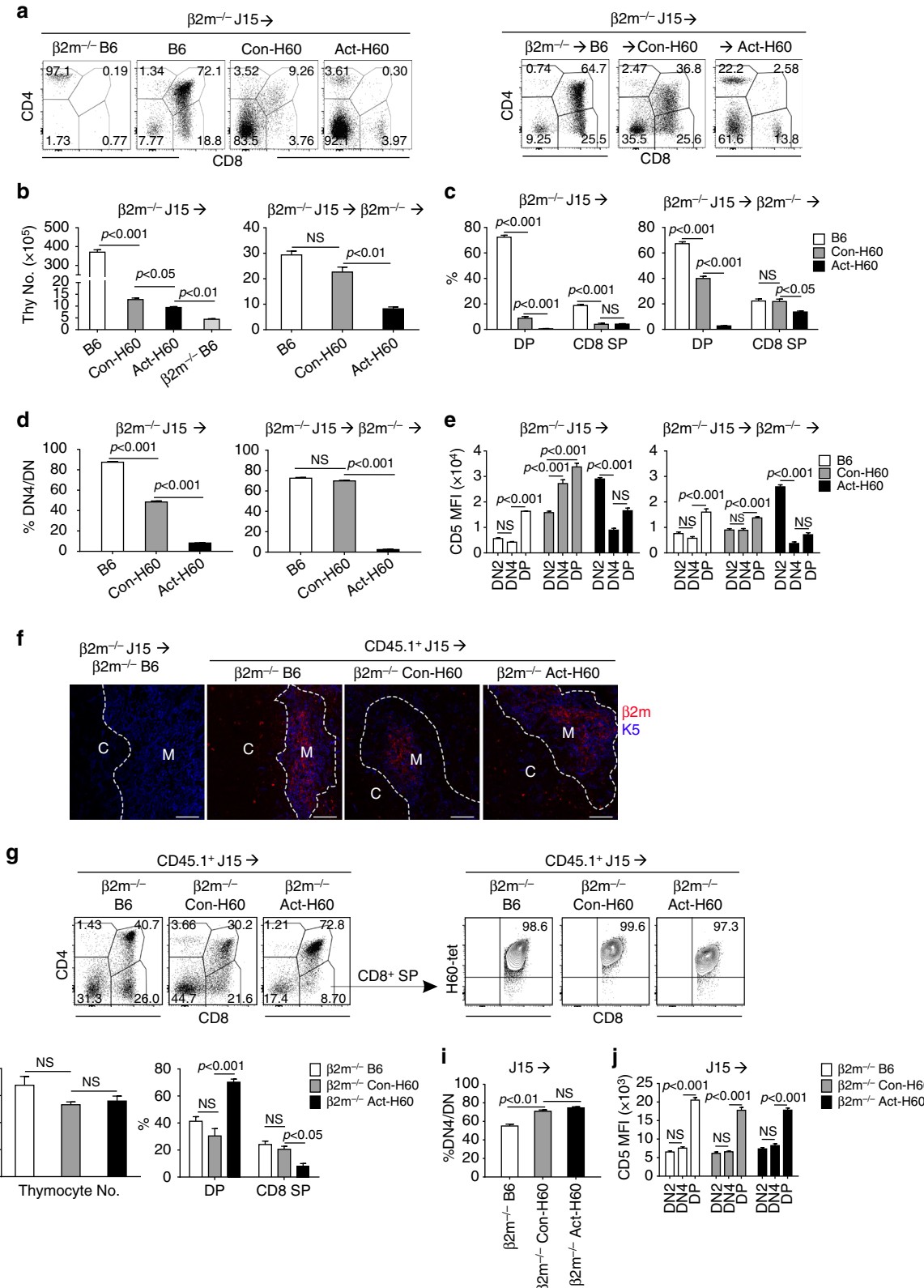

compared with H60-tetramer$^+$ CD8 T cells from B6 recipients and with H60-tetramer$^-$ cells from both recipients (Supplementary Fig. 8c–e).

In bioluminescence imaging (BLI) analysis to visualize in vivo growth of the injected tumor cells by use of luciferase-expressing H60$^+$EL4 and H60$^-$EL4 cells (H60$^+$EL4/Luc and H60$^-$EL4/Luc, respectively), luminescence signals emanated from s.c.-injected tumors were not detected since day 10 post-tumor injection in the H60$^+$EL4/Luc-injected Con-H60 and B6 recipients, while the signals increased exponentially in the other experimental groups (Fig. 6c). Thus, H60$^+$EL4 tumor cells were completely regressed in the Con-H60 recipients, confirming that the minor incidence of mortality in the Con-H60 recipients was not related to tumor growth. Collectively, these results demonstrated that J15 escapee CD8 T cells had a cytotoxic function in the periphery, contributing to the GVL effects.

**Incomplete thymic deletion of polyclonal T cells for HRA H60.** Finally, we determined whether incomplete thymic deletion also occurred in polyclonal T cells under circumstances where the frequencies of H60-cognate T cells were low in the pre-thymic repertoire like the physiological situation. To this end, CD45.1$^+$ B6 mice were used as BM donors in the H60-mismatched BMT (Fig. 7a). At 6 weeks post-BMT, enrichment of CD4-negative and H60-tetramer-binding cells by magnetic activation cell sorting (MACS) and re-staining of the MACS-enriched cells with H60-tetramers revealed that H60-cognate T cells were present in the ex vivo CD8$^+$ SP thymocytes and splenocytes from Con-H60 recipients but not in those from Act-H60 recipients (Fig. 7b). The numbers of H60-tetramer$^+$ CD8$^+$ T cells were estimated to be 302 and 72, on average, in the thymi and spleens, respectively, of Con-H60 recipients. This was significantly lower compared to the respective numbers (675 and 124) in the B6 recipients. Nonetheless, the H60-tetramer$^+$ CD8$^+$ splenic T cells generated in Con-H60 recipients were polyclonal, using diverse TCR β chains as much as did the B6 counterparts according to Shannon entropy and Simpson diversity index in TCR repertoire analysis (Supplementary Fig. 9a). Thus, incomplete thymic deletion of T cells for hematopoietic H60 was also relevant in polyclonal T cells. Similarly, significantly lower numbers of H60-tetramer$^+$ cells were identified in the ex vivo CD8$^+$ T cells isolated from normal Con-H60 mice compared to the numbers in those from normal B6 mice (Fig. 7c), confirming the incomplete thymic deletion of hematopoietic H60-specific polyclonal T cells under normal physiological conditions.

When the ex vivo CD8$^+$ thymocytes or splenocytes from Con-H60 recipients or from normal Con-H60 mice were subjected to in vitro MLC and subsequent restimulations with irradiated H60$^+$ feeder cells, H60-cognate cells could be amplified up to 2–3% of the MLC CD8 T, while the frequencies obtained with those from

B6 recipients or normal B6 mice were 5–8% (Supplementary Fig. 9b, c). H60-tetramer$^+$ CD8$^+$ T cells from Con-H60 mice were less sensitive to H60-stimulation in terms of proliferation during the MLC than were their B6 counterparts (Supplementary Fig. 9d). Thus, the lower frequency values obtained after in vitro amplification of H60-tetramer$^+$ CD8$^+$ cells from Con-H60 hosts stem from both their lower initial frequencies in vivo and slower proliferation in vitro (Supplementary Fig. 9d). The H60-tetramer$^+$ polyclonal CD8 T cells either from Con-H60 recipients after CD45.1$^+$B6 BMT or from normal Con-H60 mice were less bright in the tetramer staining compared with those from the B6 counterparts (Fig. 7b, c and Supplementary Fig. 9b, c). Thus, low avidity T cells for hematopoietic H60 escaped thymic negative selection under polyclonal conditions.

**GVL effect of polyclonal H60-cognate escapee T cells.** Tumor regression and minor GVHD-like incidences in J15 TCR Tg systems might be due to super-physiological numbers of H60-cognate escapee J15 T cells because the numbers of H60-cognate polyclonal escapee CD8 T cells were far lower than those of J15 escapee CD8 T cells in the spleens of Con-H60 recipients (i.e., 72 versus $1.5 \times 10^5$, respectively). Therefore, we wished to examine whether polyclonal H60-cognate escapee T cells would also have a tumor-eradicating capacity in the physiological BMT conditions. To this end, the three groups of polyclonal CD45.1$^+$B6 BMT recipients were injected s.c. with H60$^+$EL4 or H60$^-$EL4 cells. In the Con-H60 and B6 polyclonal recipients, H60$^+$EL4 cells regressed, while the H60$^-$EL4 cells grew exponentially. H60$^+$EL4 cell-injected Con-H60 recipients did not show any GVHD-like symptoms, including weight loss (Fig. 8a, b). Thus, H60$^+$EL4 cell-injected Con-H60 and B6 recipients showed tumor-free and disease-free long-term survival, which indicated that the polyclonal H60-cognate escapee T cells conferred resistance to the H60-positive hematological tumors.

Next, H60$^+$EL4 and H60$^-$EL4 cells expressing Thy1.1 were injected i.v. into the three different polyclonal recipients to mimic leukemic manifestation (Fig. 8c). Thy1.1$^+$ cells were detected at significantly lower frequencies in the PBLs of Thy1.1$^+$H60$^+$EL4-injected Con-H60 or B6 recipients than in the PBLs of the other experimental groups (Fig. 8d). Conversely, H60-tetramer$^+$ CD8 T cells were detected at significantly higher frequencies in the PBLs of Thy1.1$^+$H60$^+$EL4-injected Con-H60 or B6 recipients. Thus, it could be inferred that Thy1.1$^+$H60$^+$EL4 cells in the blood of Con-H60 recipients were efficiently killed by the expanded H60-cognate polyclonal escapee CD8 T cells. In the leukemic model, the death rates of H60$^+$EL4-injected polyclonal Con-H60 recipients and B6 recipients were significantly lower (mean survival time (MST) of 62 days) compared to the catastrophic deaths of the H60$^+$EL4-injected Act-H60 recipients and the H60$^-$EL4-injected recipients (MST of < 18.5 days) (Fig. 8e). This

**Fig. 4** Presentation by H60$^+$ cells in BMT recipients is responsible for negative selection of J15 thymocytes. **a–e** Flow cytometric analysis of thymocytes in recipients of $\beta 2m^{-/-}$J15 BMT. $\beta 2m^{-/-}$J15 BM cells were transplanted into B6, Con-H60, Act-H60, or $\beta 2m^{-/-}$ B6 recipients (left); or ($\beta 2m^{-/-} \rightarrow$B6), ($\beta 2m^{-/-} \rightarrow$Con-H60), or ($\beta 2m^{-/-} \rightarrow$Act-H60) chimera recipients (right). Thymocytes from the recipients were analyzed by flow cytometry 6 weeks after $\beta 2m^{-/-}$J15 BMT. **a** CD4-PE.Cy5/CD8-APC.Cy7 FACS profiles are shown after gating on H-2K$^b$ (Pacific Blue)$^-$Vβ8.3 (FITC)$^+$ cells; the percentage values of different stages are indicated. **b** The numbers of thymocytes, **c** percentages of DP and CD8$^+$ SP cells, **d** percentages of DN4 cells among the DN population, and **e** MFI values from CD5-PE staining are plotted. **f–j** CD45.1$^+$J15 BM cells were transplanted into B6, Con-H60, and Act-H60 recipients on a $\beta 2m^{-/-}$ background ($\beta 2m^{-/-}$ B6, $\beta 2m^{-/-}$ Con-H60, and $\beta 2m^{-/-}$ Act-H60). **f** Immunostaining of thymus tissue. Thymus tissue sections from recipients were stained with anti-β2m-PE and anti-keratin 5 (K5)-Alexa Fluor 647 antibodies. Images were overlaid and examined for co-localization of β2m and K5. White dotted lines are drawn around K5$^+$ cells to identify the medulla (M). C indicates the cortex. The bar indicates 100 μm. In the immunostaining procedure, $\beta 2m^{-/-}$ J15→ $\beta 2m^{-/-}$ B6 BMT was included as a negative control for β2m-staining. **g** Representative FACS profiles of CD4-PE.Cy5/CD8-APC.Cy7 staining and H60-tetramer-PE staining in CD8$^+$ SP cells are shown. **h** The numbers of thymocytes and percentages of DP and CD8$^+$ SP cells, **i** percentages of DN4 cells among the DN population, and **j** MFI values from CD5-PE staining were plotted. The data from **a–j** represent three independent experiments ($n = 3$/group/experiment) and are presented as means ± s.e.m. P-values were generated by Student's t-test

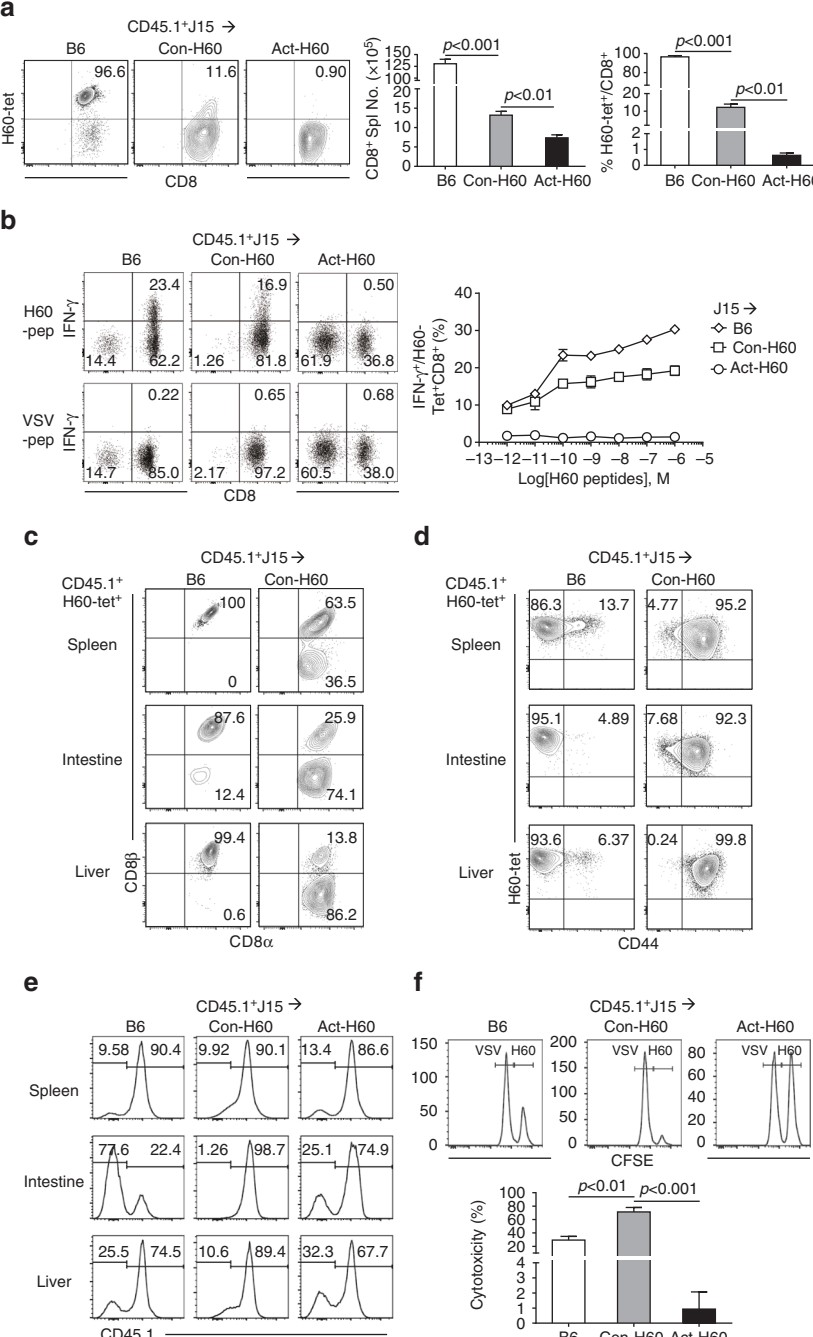

**Fig. 5** Escape from thymic deletion of lower avidity T cells and their effector differentiation. **a** Analysis of splenic CD8[+] T cells from CD45.1[+]J15 BMT recipients. The numbers of splenic CD8[+] T cells and the frequencies of H60-tetramer-binding cells among the splenic CD8 T cells are plotted. Representative FACS data show H60-tetramer-PE staining in splenic CD8 T cells after gating on CD45.1[+] cells; the percentages of H60-tetramer-binding cells are indicated. **b** TCR avidity assay with intracytoplasmic IFN-γ-staining. Splenic CD8 T cells from three different recipients were stimulated in vitro in the presence of different concentrations of H60 or VSV peptides and then subjected to intracytoplasmic anti-IFN-γ staining. **c–e** Flow cytometric analysis of the leukocytes infiltrating the spleen, intestines, and liver in the recipients of CD45.1[+]J15 BMT at 6 weeks post-BMT. **c** CD8β-PE.Cy7 by CD8α-APC.Cy7 and **d** H60-tetramer-PE by CD44-APC profiles of the CD45.1[+]H60-tetramer[+]CD8α[+] cells are shown; the percentage values of each quadrat are indicated. **e** Representative single histograms indicating the origin of leukocytes based on CD45.1 expression. Percentage values of CD45.1[−] cells (recipient origin) and CD45.1[+] cells (donor origin) are indicated. **f** In vivo cytotoxicity assay. A 1:1 mixture of CFSE-labeled and H60 (CFSE[hi]) and VSV (CFSE[low]) peptide-loaded target cells was injected into CD45.1[+]J15 BMT recipients. A representative single histogram of PBLs from the recipients is shown for the presence or absence of CFSE-labeled target cells. In vivo cytotoxicity was expressed as the percentage of specific lysis, calculated by the equation (1–(% H60-loaded targets/% VSV-loaded targets)) × 100. Percentage cytotoxicity values are plotted. All data (**a–f**) represent more than three independent experiments (n = 3/group/experiment) and are presented as means ± s.e.m. *P*-values were generated by Student's *t*-test

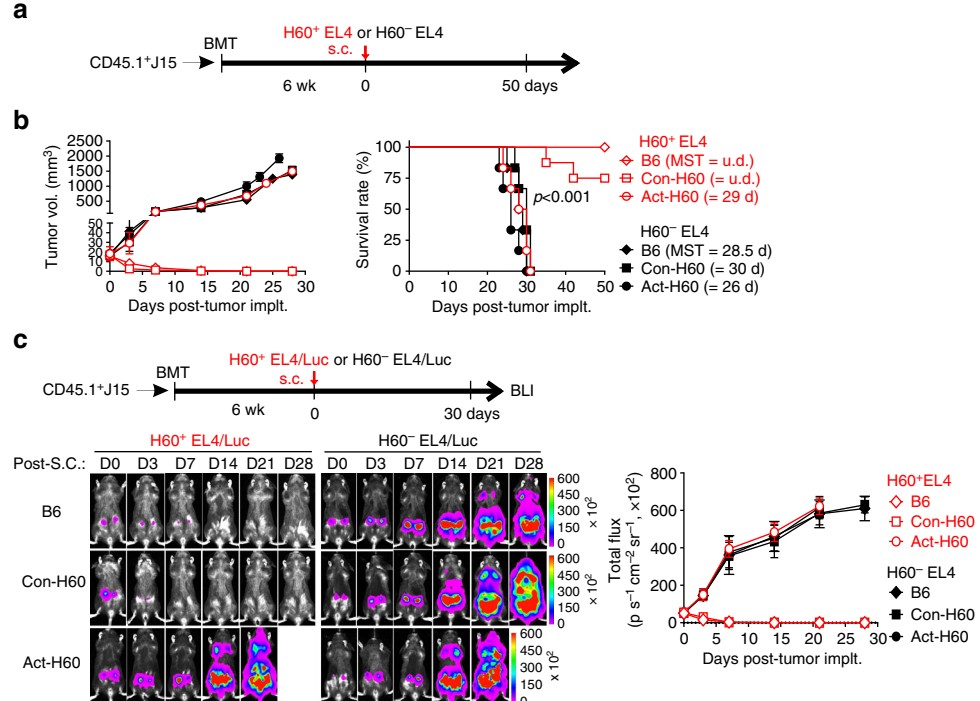

**Fig. 6** Protection from H60+EL4 tumor cells by J15 CD8+ escapee T cells in Con-H60 recipients. **a** Schematic presentation of the experimental procedure. H60+EL4 or H60−EL4 tumor cells ($5 \times 10^5$) were injected s.c. into the recipients of CD45.1+J15 BMT at 6 weeks after BMT. **b** The tumor volume and survival rate of CD45.1+J15 BMT recipients were periodically measured and plotted. The mean survival times (MST) of tumor-implanted recipients are indicated in parentheses. Data represent two independent experiments (n = 6 mice/group/experiment). P-values were determined by the log-rank (Mantel–Cox) test. **c** In vivo BLI of tumor-injected recipients. Recipients of CD45.1+J15 BMT after s.c. injection of $5 \times 10^5$ H60+EL4-Luc or H60−EL4-Luc cells were imaged for luminescence signal emission from the tumor cells on days 0, 3, 7, 14, 21, and 28 post-tumor implantation. The emitted luminescence signal values are plotted. Representative data from two independent experiments (n = 3/group/experiment) are shown

GVL effect was recapitulated in another leukemia model using a myeloid leukemia cell line, C1498. When Thy1.1+H60+ or H60−C1498 cells were injected i.v. into polyclonal recipients (Fig. 8c), only the B6 or Con-H60 recipients with Thy1.1+H60+C1498 survived for a long period: H60-tetramer+ CD8 T cells were detected at significant frequencies in peripheral blood, while Thy1.1+ tumor cells were rarely detected (Fig. 8f, g). None of the Con-H60 polyclonal recipients in the EL4 or C1498 leukemia model showed GVHD-like symptoms. Taken together, these results demonstrate that hematopoietic H60-specific polyclonal escapee CD8 T cells contribute to GVL effects in the periphery.

## Discussion

Antigen distribution patterns influence the extent of T-cell-negative selection in the thymus. However, whether and how hematopoietic cell-restricted distribution impacts thymic deletion and peripheral immunity have not been addressed in detail to date. Using a natural hematopoietic antigen, MiHA H60, we demonstrated that thymic negative selection is not strict to the T cells specific for HRAs, with low avidity T cells escaping deletion, and the deletion escapee T cells in the periphery conferring resistance to hematological tumors. This study is the first to demonstrate incomplete thymic negative selection of HRA-specific CD8 T cells and mediation of GVL effects by HRA-specific deletion escapees.

Our finding that T cells specific for the antigen expressed by thymic DCs are incompletely deleted in the thymus is somewhat unexpected and differs from the conventional knowledge that DCs are major and efficient APCs negatively selecting

thymocytes[12–14]. Partial thymic deletion has been demonstrated regarding the CD4 or CD8 T cells of which cognate antigens are expressed in a restricted manner by peripheral tissues and mTECs in the thymus[8–10,41,47]. However, none of the previous studies focused specifically on the thymic negative selection of HRA-specific T cells. One report described a partial deletion of eGFP/eYFP-specific CD4 T cells in mice where the fluorescence proteins were expressed by epidermal DCs and some thymic DCs only under the control of the langerin promoter[10]. However, this report also showed strict thymic negative selection of the CD4 T cells in mice where the fluorescence proteins were expressed by all the thymic DCs under the control of the CD11c promoter. The authors suggested that the difference in the numbers of APCs presenting cognate peptide-MHC complexes in the thymus (>100-fold higher in the case of CD11c than in that of langerin) might cause the different levels of thymic deletion. This APC-abundance hypothesis can also explain the partial versus complete deletion of H60-cognate thymocytes in the Con-H60 (with DCs as the only APCs) versus Act-H60 thymus (with DCs and stromal APCs), respectively. However, the fact that all the thymic DCs of the Con-H60 mouse express H60 indicates that the abundance of cognate APCs is not the sole factor influencing the levels of thymic deletion, and the levels of peptide/MHC complexes displayed on thymic DCs are an additional important factor. The number of H60/H-2K$^b$ complexes has been estimated to be 5–15 copies per cell on the H60+ tumor cells[48]. We consider that the natural levels of H60/H-2K$^b$ complexes on Con-H60 thymic DCs around this approximated number would not sufficiently delete the H60-cognate thymocytes, including those with low avidity, and suggest that complete thymic negative selection

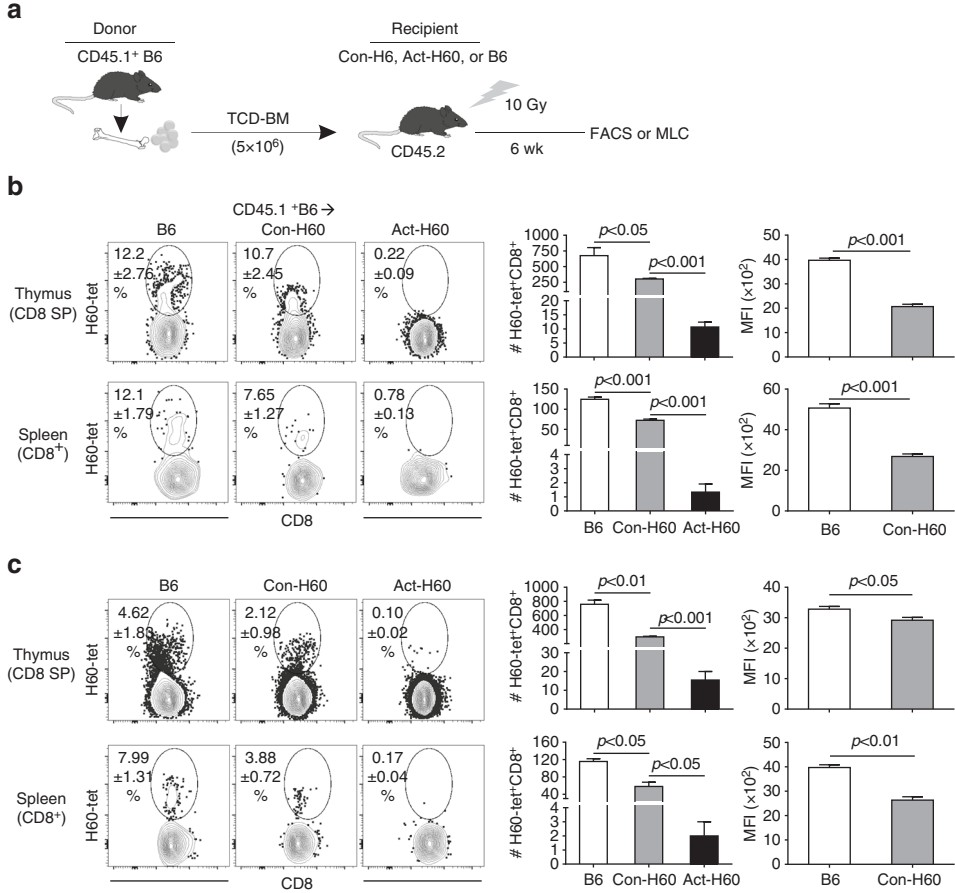

**Fig. 7** Escape from thymic deletion of polyclonal CD8⁺ T cells specific for hematopoietic H60. **a** A schematic illustration depicting the experimental design. TCD-BM cells (5 × 10⁶) from CD45.1⁺B6 mice were transplanted into B6, Con-H60, and Act-H60 mice that were lethally irradiated. Six weeks after BMT, CD4 cells MACS-purified from the thymocytes and splenocytes of the recipients were subjected to flow cytometric analysis or MLC. **b** Detection of H60-cognate cells ex vivo from BMT recipients. CD4-negative and H60-tetramer-binding cells were enriched via MACS from the thymocytes and splenocytes of the recipients and were stained with H60-tetramer-PE, anti-CD45.1-FITC, and anti-CD8-allophycocyanin antibodies. Representative FACS profiles are shown after CD45.1⁺ cell gating. The values in the FACS profiles indicate the percentages of H60-tetramer⁺ cells among the CD45.1⁺ CD8⁺ cells. The numbers of H60-cognate CD8 T cells (estimated based on the FACS data) and MFI values from H60-tetramer staining are presented as bar graphs. **c** Detection of H60-cognate cells ex vivo from normal mice. The CD4⁻H60-tetramer⁺ cells were enriched from the B6, Con-H60, and Act-H60 mice at 8 weeks old, as described above, and stained with H60-tetramer-PE, anti-CD3ε-FITC, and anti-CD8-allophycocyanin antibodies. Representative FACS profiles are shown after CD3ε⁺ cell gating; the percentages of H60-tetramer⁺ cells are indicated. Estimated numbers of H60-cognate CD8 T cells and MFI values of H60-tetramer staining are plotted. The data from **b**, **c** represent three independent experiments (n = 2/group/experiment) and are presented as means ± s.e.m. P-values were generated by Student's t-test

of HRA-specific T cells may be infrequent events under natural and physiological conditions.

H60-specific escapee CD8 T cells generated in the periphery have effector differentiation potentials. Their cytotoxic effector potentials were manifested by the regression of H60⁺ hematological tumors, especially under the circumstances of the hematopoietic H60-mismatched BMT. Mediation of GVL effects by the HRA-specific escapee CD8 effector cells is supported not only by the H60⁺ tumor regression but also by their expansion and enhanced donor leukocyte chimerism in the blood of the Con-H60 recipients. These blood profiles mimic those observed in leukemia patients who received allo-BMT and showed a favorable clinical course[49,50]. However, the fact that the majority of J15→Con-H60 BMT recipients, and all of the (J15×Con-H60) F1 progenies and their polyclonal counterparts were healthy suggests that the effector potentials of H60-specific escapee CD8 T cells would be irrelevant to autoimmune induction, in spite of the fact that opportunities for the H60-specific escapee CD8 T cells to encounter H60⁺ hematopoietic cells would be numerous in multiple peripheral organs. In the case of TRA-specific escapee

CD8 T cells, their autoimmune effector potentials were apparent after antigen priming under inflammatory conditions[8,9]. However, escapee CD4 T cells specific for some TRAs demonstrated low effector potentials[10]. Thus, it is likely that the effector potential and fate in the periphery differ between CD4 and CD8 escapee T cells and also vary according to the tissue distribution of the cognate antigens.

Clinical studies have demonstrated the association of hematopoietic MiHA-mismatched allo-BMT with high incidences of GVHD[51,52] or GVL effects only in the presence of GVHD[53,54]. The main GVHD inducers in those studies were mature donor T cells present in the BM inoculum and then activated in the acute phase following allo-BMT. However, in our study, pathological incidence was low, locally restricted to the intestine, and observed only in the Con-H60 recipients of J15 BM, and not B6 BM. Thus, the few occasions of intestine-like GVHD in the Con-H60 BMT recipients are ascribed to the non-physiologically high frequencies and numbers of H60-cognate escapee CD8 T cells in the periphery due to the transgenic design of the J15 T cell development. In addition, inflammatory signals generated due to

lethal irradiation and the intestinal tissue environment play roles in enhancing the infiltration and effector differentiation of the H60-cognate escapee CD8 T cells specifically in the intestine[55–57]. However, polyclonal escapee T cells did not induce GVHD but retained their anti-tumor reactivity. This highlights the value of thymus-derived HRA-specific T cells compared with mature

donor HRA-specific T cells in GVL without GVHD, although the application to other HRAs and the clinical translation need to be further investigated.

In summary, we demonstrated that CD8 T cells specific for hematopoietic MiHA can evade full thymic deletion, and the deletion escapee CD8 T cells have effector differentiation

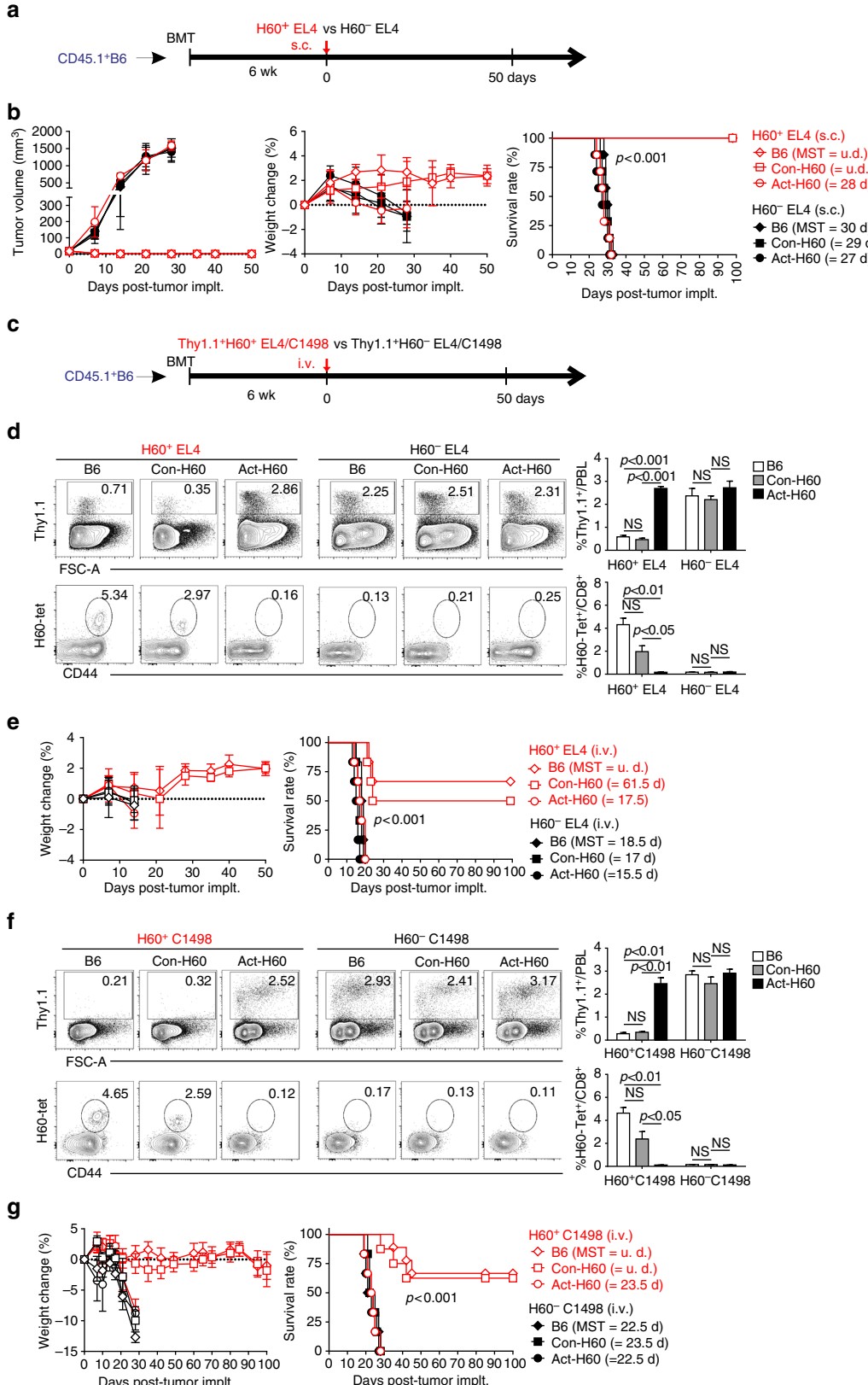

potentials that can mediate GVL effects in the recipients of hematopoietic antigen-mismatched BMT. These results add valuable information to our understanding of the immunological implications of HRAs and the clinical benefits of hematopoietic MiHA-mismatched BMT.

## Methods

**Mice.** The B6, B6.SJL-$Ptprc^aPep3^b$/BoyJ (CD45.1$^+$), B6.129P2-$B2m^{tm1Unc}$ ($\beta2m^{-/-}$), and B6.129S7-$Rag-1^{tm1Mom}$/J ($Rag-1^{-/-}$) mouse strains were purchased from The Jackson Laboratory (Bar Harbor, ME, USA). B6.Tg(H60a)114/DCR (Act-H60) and B6.C-$H60^c$/DCR (Con-H60) were kind gifts from Dr. Derry Roopenian (The Jackson Laboratory). The TCR transgenic (B6.Tg(TCRaTcrb)J15/EYC) mouse strain was previously described[32] and backcrossed onto the CD45.1$^+$, $\beta2m^{-/-}$, or $Rag-1^{-/-}$ background. Mice were maintained under specific pathogen-free conditions at the Biomedical Center for Animal Resource Development of the Seoul National University College of Medicine. Male or female mice were used between 8 and 12 weeks of age. All experiments were performed after approval from the Seoul National University Institutional Animal Care and Use Committee was obtained and in accordance with the guidelines.

**Bone marrow transplantation and tumor injection.** Recipient mice were irradiated with two split doses of 1,000 cGy from a $^{137}$Cs source (IBL 437 C; CIS Bio International, Bangnols sur Ceze, France) with a 5-h interval. BM cells isolated from the femur and tibia of gender-matched donors were depleted of T cells by MACS and $5 \times 10^6$ T cell-depleted BM (TCD-BM) cells were transferred i.v. into recipients 5 h after the second irradiation as previously described[57,58]. EL4 cells were purchased from ATCC (TIB-39; Rockville, MD, USA), and transduced to express H60, H60 and Luciferase, or H60 and Thy1.1 (H60$^+$EL4) or transduced with empty vector[32]. Original and transduced EL4 cells were periodically checked for expression of CD3 and the relevant antigens by flow cytometry. H60-positive or negative EL4 tumor cells were injected s.c. ($5 \times 10^5$) or i.v. ($1 \times 10^6$) into BMT recipients at 6 weeks after BMT. C1498 (B6 acute myeloid leukemia cells) were purchased from ATCC (TIB-49) and transduced to express Thy1.1 alone or together with H60 as described above. Original and transduced C1498 cells were periodically checked for expression of the relevant antigens by flow cytometry. H60-positive or -negative C1498 leukemic cells were injected i.v. ($5 \times 10^5$) into BMT recipients at 6 weeks after BMT. Both original and transduced EL4 and C1498 cell lines have been periodically tested to exclude mycoplasma contamination by PCR.

**Antibodies and flow cytometric analysis.** Fluorescein isothiocyanate (FITC)-conjugated, phycoerythrin (PE)-conjugated, allophycocyanin-conjugated, Pacific Blue-conjugated, or CyChrome (Cy)-conjugated monoclonal antibodies against CD3ε (145-2C11), CD4 (GK1.5), CD5 (53-7.3) CD8α (53-6.7), CD24 (M1/69), CD25 (PC61), CD44 (IM-7), CD45.1 (A20), CD69 (H1.2F3), TCR β (H57-597), TCR Vβ8.3 (1B3.3), H2-Kb (AF6-88.5), β2m (B10.S), α4β7 (DATK32), and CXCR6 (SA051D1) were purchased from BioLegend (San Diego, CA, USA), BD Biosciences (San Jose, CA, USA), eBioscience (San Diego, CA, USA), Santa Cruz Biotechnology (Santa Cruz, CA, USA), and Invitrogen (Carlsbad, CA, USA). Ex vivo or in vitro cells were stained with antibodies and/or H60-tetramers (LTFNYRNL/H-2K$^b$) in staining buffer ($1 \times$ phosphate-buffered saline (PBS) with 0.1% bovine calf serum and 0.05% sodium azide) at 4 °C for 30 min and analyzed using FACS LSRII (BD Biosciences) and FlowJo software (TreeStar, Ashland, OR, USA). Flow cytometry gating strategies are shown in Supplementary Fig. 1 for analysis of thymocytes and splenocytes, and in Supplementary Fig. 5b for sorting of thymic DCs and mTECs.

**TCR avidity assay with intracytoplasmic IFN-γ staining.** One million splenocytes from J15 BMT recipients or J15 F1 mice were incubated in the presence of the indicated concentrations of H60 (LTFNYRNL) or VSV (RGYVYQGL) peptides in 96-well round bottom plates in Dulbecco's modified Eagle's Medium (DMEM; Gibco, Carlsbad, CA, USA) for 4 h. Brefeldin A (Sigma-Aldrich, St. Louis, MO, USA) was added to final concentration of 10 μg/ml for last 2 h. Then, cells were incubated with ethidium monoazide bromide (5 μg/ml; Invitrogen) and surface-stained with fluorescence dye-conjugated antibodies after washing. After washing and fixation with 1 % paraformaldehyde in PBS, cells were stained with anti-IFN-γ-PE antibody diluted in PBS containing 0.1% saponin (Sigma-Aldrich) for flow cytometric analysis, as described previously[31].

**Staining and MLC of MACS-enriched cells.** CD4$^-$ cells that had been MACS-purified from thymocytes and splenocytes were incubated with PE-labeled H60-tetramers and then with anti-PE antibody-conjugated magnetic beads (Miltenyi Biotech, Auburn, CA, USA) for enrichment of the antigen-specific T cells as described previously[32,59,60]. These MACS-enriched cells were stained with H60-tetramers and antibodies for flow cytometric analysis. Alternatively, the CD4$^-$ cells were subjected to MLC for 7 days and then re-stimulated weekly with irradiated (2000 cGy) Con-H60 splenocyte feeder cells in DMEM containing 5% FBS (Hyclone, Logan, UT, USA) and rIL-2 (50 U/ml; Sigma-Aldrich) as previously described[27].

**In vivo cytotoxicity.** B6 splenocytes pulsed with H60 or VSV peptide were labeled with 2.5 or 0.25 μM CFSE (carboxyfluorescein succinimidyl ester; Invitrogen), respectively, injected i.v. into the BMT recipients, and detected 72 h later by flow cytometry of the PBLs from the recipients as previously described[31].

**BrdU incorporation assay.** During the MLC of CD8$^+$ thymocytes or splenocytes from BMT recipients or normal mice with irradiated H60$^+$ feeder cells, BrdU (10 μM; Sigma-Aldrich) was first added to each culture-medium-containing well of a 24-well plate at 24 h post-MLC and supplemented at every 12 h afterwards. The cells were collected at the indicated time points post-BrdU treatment. After washing, the collected cells were fixed, permeabilized, and treated with DNase I (Sigma-Aldrich) in 0.15 M NaCl/4.2 mM MgCl$_2$ prior to staining with anti-BrdU antibody (B44; BD Biosciences) as described previously[61].

**Immunofluorescence staining of the thymus.** Snap frozen thymus tissue serially cut at 10 μm intervals was fixed with 4% paraformaldehyde, washed with $1 \times$ PBS, and incubated with blocking buffer (5% normal goat serum and 0.3% Triton X-100). Next, the tissue was stained with a mixture of unconjugated anti-keratin 5 (AF138; Covance, Princeton, NJ, USA) and PE-conjugated anti-β2m (B10.S; Santa Cruz Biotechnology) antibodies overnight at 4 °C. After washing with $1 \times$ PBS, the tissue was stained with secondary antibodies for 2 h, followed by staining with 4′6′-diamidino-2-phenylindole (DAPI) (Invitrogen). Anti-keratin 5 was vitalized by goat anti–rabbit IgG Alexa Fluor 647 (Invitrogen). The fluorophores were excited by laser at 405, 488, 515, and 633 nm and detected by a scanning confocal microscope (FV-1000; Olympus, Tokyo, Japan). Images were analyzed using FV10-ASW Fluoview software (Olympus), ImageJ (National Institutes of Health, Bethesda, MD, USA), and Adobe Photoshop (Adobe Systems Inc., Mountain View, CA, USA) software.

**Histological examination.** Peripheral organs obtained from BMT recipients at 8 weeks post-BMT were fixed in 4% formaldehyde solution, cut transversely, embedded in paraffin, and serially sectioned. Tissue sections were stained with hematoxylin and eosin (H&E) for analysis of general morphology. The

**Fig. 8** Protection from hematological tumors by H60-specific polyclonal CD8$^+$ escapee T cells. **a** Schematic presentation of the experimental procedure. H60$^+$EL4 or H60$^-$EL4 tumor cells ($5 \times 10^5$) were injected s.c. into the recipients at 6 weeks post-CD45.1$^+$B6 BMT and observed on a weekly basis. **b** The tumor volumes, weight changes, and survival rates of the recipients were plotted. Data are representative of two independent experiments ($n = 7$/group/experiment). The MSTs of tumor-implanted recipients are indicated in parentheses. $P$-value was determined by the log-rank (Mantel–Cox) test. **c** Experimental scheme. Thy1.1$^+$H60$^+$EL4 vs Thy1.1$^+$H60$^-$EL4 tumor cells ($1 \times 10^6$) or Thy1.1$^+$H60$^+$C1498 vs Thy1.1$^+$H60$^-$C1498 acute myeloid leukemia cells ($5 \times 10^5$) were injected i.v. into the recipients at 6 weeks post-CD45.1$^+$B6 BMT. Blood was collected on a weekly basis for flow cytometric analysis. **d** Flow cytometric analysis of PBLs from EL4 tumor-injected recipients for the detection of Thy1.1$^+$ tumor cells and H60-tetramer$^+$ CD8 T cells in blood. Representative FACS data show the detection of Thy1.1$^+$ cells in live cells and H60-tetramer$^+$ cells in CD8$^+$ gated cells in the blood of recipients on day 21 post-tumor injection. The percentages of Thy1.1$^+$ cells in PBL or H60-tetramer$^+$ cells in CD8 cells are indicated in the FACS data and are plotted. The data (**d**) represent three independent experiments ($n = 3$/group/experiment) and are presented as means ± s.e.m. $P$-values were generated by the Student's $t$-test. **e** Weight changes and survival rates of BMT recipients injected with Thy1.1$^+$H60$^+$EL4 or Thy1.1$^+$H60$^-$EL4 cells. Data representative of two independent experiments ($n = 6$/group/experiment) are shown. The MSTs of tumor-injected recipients are indicated in parentheses. $P$-value was determined by the log-rank (Mantel–Cox) test. **f** Flow cytometric analysis of PBLs, and **g** weight changes and survival rates of C1498 tumor-injected recipients. Data (**f**, **g**) were processed as described for **d**, **e**. Data are representative of two (**g**; $n = 8$/group/experiment) or three (**f**; $n = 3$/group/experiment) independent experiments

histopathological changes of the small and large intestines were assessed and scored based on crypt apoptosis and inflammation, as described previously[62].

**BLI**. BMT recipients implanted with H60+EL4/Luc or H60− EL4/Luc cells were imaged under anesthetization[63] using the Kodak In Vivo Multispectral Imaging System FX (Carestream Health, Rochester, NY, USA) for 5 min after injection of 100 μl D-luciferin (150 mg/kg body weight; Caliper Life Sciences, Hopkinton, MA, USA). Images were processed and analyzed using the Kodak imaging software (Carestream Health).

**RT–PCR**. Total RNA was extracted from FACS-sorted mTECs, thymic DCs, or intestine-infiltrating leukocytes. PCR was performed after cDNA synthesis using oligo dT primers. The oligonucleotide primer sequences used for PCR were as follows: H60 forward, 5′-TTCCTCATCACATATTTCAGTCAC-3′; H60 reverse, 5′-TGACACTCAGACCCTGGTTGTCAG-3′; β-actin forward, 5′-GGCTGTATTCCCCTCCATCG-3′; β-actin reverse, 5′-CCAGTTGGTAA-CAATGCCATGT-3′. The thermal cycling conditions were an initial denaturation step at 95 °C for 3 min and 40 cycles of 95 °C for 30 s, 55 °C for 30 s and 72 °C for 30 s.

**TCR repertoire analysis**. Total RNA was extracted from sorted cells with RNeasy Plus Universal Kit (Qiagen, Hilden, Germany) according to the manufacturer's instructions. Next-generation sequencing was performed with an unbiased TCR repertoire analysis technology developed by Repertoire Genesis Inc. (Osaka, Japan). An unbiased adaptor-ligation PCR was performed according to the previous report[64]. In brief, total RNA was converted to complementary DNA (cDNA) with Superscript III reverse transcriptase (Invitrogen). Then, double strand (ds)-cDNA was synthesized and an adaptor was ligated to the 5′ end of the ds-cDNA and then cut with SphI restriction enzyme. PCR was performed with P20EA adaptor primer (5′-TAATACGACTCCGAATTCCC-3′) and TCR β-chain constant region-specific (mCB1, 5′-AGGATTGTGCCAGAAGGTAG-3′). The PCR conditions were as follows: 95 °C (20 s), 65 °C (30 s), and 72 °C (1 min) for 20 cycles. The second PCR was performed with mCB2 (5′-TTGTAGGCCTGAGGGTCC-3′) and P20EA primers under the same PCR conditions. After Tag PCR amplification, index (bar-code) sequences were added by amplification with Nextera XT index kit v2 setA (Illumina, San Diego, CA). Sequence was done with the Illumina Miseq paired-end platform (2 × 300 bp). Data processing, assignment, and data aggregation were automatically performed using repertoire analysis software originally developed by Repertoire Genesis, Inc. TCR sequences was assigned with a data set of reference sequences from the international ImMunoGeneTics information system (IMGT) database (http://www.imgt.org). Nucleotide sequences of CDR3 regions ranged from conserved Cysteine at position 104 (Cys104) of IMGT nomenclature to conserved tryptophan (Trp118) or phenylalanine at position 118 (Phe118) and the following glycine (Gly119) were translated to deduced amino acid sequences. A unique sequence read (USR) was defined as a sequence read having no identity in TRV, TRJ, and deduced amino acid sequence of CDR3 with the other sequence reads. The copy number of identical USR were automatically counted by Repertoire Genesis software. After removal of sequences with low quality scores, TCR repertoire analysis was performed using bioinformatics software created by Repertoire Genesis Incorporation.

**Statistical analysis**. Prism (GraphPad Software Inc., La Jolla, CA, USA) was used for the statistical analyses. Kaplan–Meier survival curves were compared using log-rank tests. Data are presented as means ± s.e.m. P-values were determined by unpaired Student's t-tests. A P-value <0.05 was considered to indicate statistical significance.

**Data availability**. Sequence data that support the findings of this study have been deposited in Sequence Read Archive with the primary accession code SRP125708.

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

# ARTICLE

33. Jeon, J. Y., Jung, K. M., Chang, J. & Choi, E. Y. Characterization of CTL clones specific for single antigen, H60 minor histocompatibility antigen. *Immune Netw.* **11**, 100–106 (2011).

34. Li, N. et al. Memory T cells from minor histocompatibility antigen-vaccinated and virus-immune donors improve GVL and immune reconstitution. *Blood* **118**, 5965–5976 (2011).

35. Schonrich, G. et al. Down-regulation of T cell receptors on self-reactive T cells as a novel mechanism for extrathymic tolerance induction. *Cell* **65**, 293–304 (1991).

36. Rocha, B. & von Boehmer, H. Peripheral selection of the T cell repertoire. *Science* **251**, 1225–1228 (1991).

37. Liu, G. Y. et al. Low avidity recognition of self-antigen by T cells permits escape from central tolerance. *Immunity* **3**, 407–415 (1995).

38. Ju, J.-M. et al. Selection of thymocytes expressing transgenic TCR specific for a minor histocompatibility antigen, H60. *Immune Netw.* **15**, 222–231 (2015).

39. Azzam, H. S. et al. CD5 expression is developmentally regulated by T cell receptor (TCR) signals and TCR avidity. *J. Exp. Med.* **188**, 2301–2311 (1998).

40. McCaughtry, T. M., Baldwin, T. A., Wilken, M. S. & Hogquist, K. A. Clonal deletion of thymocytes can occur in the cortex with no involvement of the medulla. *J. Exp. Med.* **205**, 2575–2584 (2008).

41. Gallegos, A. M. & Bevan, M. J. Central tolerance to tissue-specific antigens mediated by direct and indirect antigen presentation. *J. Exp. Med.* **200**, 1039–1049 (2004).

42. Hubert, F. X. et al. Aire regulates the transfer of antigen from mTECs to dendritic cells for induction of thymic tolerance. *Blood* **118**, 2462–2472 (2011).

43. Pobezinsky, L. A. et al. Clonal deletion and the fate of autoreactive thymocytes that survive negative selection. *Nat. Immunol.* **13**, 569–578 (2012).

44. Berlin, C. et al. Alpha 4 beta 7 integrin mediates lymphocyte binding to the mucosal vascular addressin MAdCAM-1. *Cell* **74**, 185–195 (1993).

45. Hamann, A., Andrew, D. P., Jablonski-Westrich, D., Holzmann, B. & Butcher, E. C. Role of alpha 4–integrins in lymphocyte homing to mucosal tissues in vivo. *J. Immunol.* **152**, 3282–3293 (1994).

46. Sato, T. et al. Role for CXCR6 in recruitment of activated CD8+lymphocytes to inflamed liver. *J. Immunol.* **174**, 277–283 (2005).

47. Lucca, L. E. et al. Myelin oligodendrocyte glycoprotein induces incomplete tolerance of CD4(+) T cells specific for both a myelin and a neuronal self-antigen in mice. *Eur. J. Immunol.* **46**, 2247–2259 (2016).

48. Malarkannan, S. et al. The molecular and functional characterization of a dominant minor H antigen, H60. *J. Immunol.* **161**, 3501–3509 (1998).

49. Kroger, N. et al. NCI First International Workshop on the Biology, Prevention, and Treatment of Relapse after Allogeneic Hematopoietic Stem Cell Transplantation: report from the Committee on Disease-Specific Methods and Strategies for Monitoring Relapse following Allogeneic Stem Cell Transplantation. Part I: Methods, acute leukemias, and myelodysplastic syndromes. *Biol. Blood Marrow Transplant.* **16**, 1187–1211 (2010).

50. van der Torren, C. R. et al. Possible role of minor H antigens in the persistence of donor chimerism after stem cell transplantation; relevance for sustained leukemia remission. *PLoS ONE* **10**, e0119595 (2015).

51. Goulmy, E. et al. Mismatches of minor histocompatibility antigens between HLA-identical donors and recipients and the development of graft-versus-host disease after bone marrow transplantation. *N. Engl. J. Med.* **334**, 281–285 (1996).

52. Gallardo, D. et al. Disparity for the minor histocompatibility antigen HA-1 is associated with an increased risk of acute graft-versus-host disease (GvHD) but it does not affect chronic GvHD incidence, disease-free survival or overall survival after allogeneic human leucocyte antigen-identical sibling donor transplantation. *Br. J. Haematol.* **114**, 931–936 (2001).

53. Mutis, T. et al. Graft-versus-host driven graft-versus-leukemia effect of minor histocompatibility antigen HA-1 in chronic myeloid leukemia patients. *Leukemia* **24**, 1388–1392 (2010).

54. Spierings, E. et al. Multicenter analyses demonstrate significant clinical effects of minor histocompatibility antigens on GvHD and GvL after HLA-matched related and unrelated hematopoietic stem cell transplantation. *Biol. Blood Marrow Transplant.* **19**, 1244–1253 (2013).

55. Obar, J. J. et al. Pathogen-induced inflammatory environment controls effector and memory CD8+T cell differentiation. *J. Immunol.* **187**, 4967–4978 (2011).

56. Plumlee, C. R., Sheridan, B. S., Cicek, B. B. & Lefrancois, L. Environmental cues dictate the fate of individual CD8+T cells responding to infection. *Immunity* **39**, 347–356 (2013).

57. Lim, J. Y. et al. MyD88 in donor bone marrow cells is critical for protection from acute intestinal graft-vs.-host disease. *Mucosal Immunol.* **9**, 730–743 (2016).

58. Ju, J.-M., Lee, H., Oh, K., Lee, D.-S. & Choi, E. Y. Kinetics of IFN-γ and IL-17 production by CD4 and CD8 T cells during acute graft-versus-host disease. *Immune Netw.* **14**, 89–99 (2014).

59. Moon, J. J. et al. Naive CD4(+) T cell frequency varies for different epitopes and predicts repertoire diversity and response magnitude. *Immunity* **27**, 203–213 (2007).

60. Obar, J. J., Khanna, K. M. & Lefrançois, L. Endogenous naive CD8+T cell precursor frequency regulates primary and memory responses to infection. *Immunity* **28**, 859–869 (2008).

61. Anderson, S. M. et al. Taking advantage: high-affinity B cells in the germinal center have lower death rates, but similar rates of division, compared to low-affinity cells. *J. Immunol.* **183**, 7314–7325 (2009).

62. Kaplan, D. H. et al. Target antigens determine graft-versus-host disease phenotype. *J. Immunol.* **173**, 5467–5475 (2004).

63. Song, M. G. et al. In vivo imaging of differences in early donor cell proliferation in graft-versus-host disease hosts with different pre-conditioning doses. *Mol. Cell* **33**, 79–86 (2012).

64. Yoshida, R. et al. A new method for quantitative analysis of the mouse T-cell receptor V region repertoires: comparison of repertoires among strains. *Immunogenetics* **52**, 35–45 (2000).

## Acknowledgements

This study was supported by grants from Seoul National University Hospital, the National Research Foundation funded by the Ministry of Science and ICT (Basic Science Research Program 2016R1A2B3016080; Basic Research Laboratory 2017R1A4A1015745) and the Korea Healthcare Technology R&D Project, Ministry of Health and Welfare (HI16C0047), Republic of Korea.

## Author contributions

J.-M.J. performed the experiments and wrote the manuscript; M.H.J., G.N., W.K., S.O., and. H.D.K. performed the experiments; J.Y.K. and J.C. designed the experiments and provided the tetramers; S.H.L. and G.S.P. performed the blind histologic assessment; C.-K.M., D.-S.L., K.C., and M.G.K. contributed to the experimental design and data interpretation; E.Y.C. contributed to the experimental design and data interpretation, and wrote the manuscript; and all the authors reviewed and edited the manuscript.

## Additional information

**Competing interests:** The authors declare no competing financial interests.

