## [Peer review file · Nature Communications]

Reviewers' comments:

Reviewer #1 (Remarks to the Author):

This study examines thymic selection against the minor histocompatibility antigen H60. The authors rightly indicate that while there has been a greater understanding of how the thymus tolerises against tissue restricted antigens expressed by medullary thymic epithelium, tolerance to antigens that may be expressed exclusively by haemopoietic cells is less well understood.

The authors provide strong data that H60-specific T-cells escape thymic deletion, and that these cells can mediate GVL effects in the periphery. However, what is much less clear is the nature of the intrathymic cell that is expressing H60 to impose tolerance. The model proposed relies heavily on the notion that H60 is expressed exclusively by haemopoietic cells and not medullary thymic epithelium.

Given that many so-called tissue restricted antigens are now known to be expressed by mTEC, can the authors provide clear evidence that H60 is expressed only by haemopoietic cells, and be absent from mTEC? If not, then the data is compatible with the notion that mTEC may be mediating tolerance, rather than haemopoietic cells.

Reviewer #2 (Remarks to the Author):

The authors address the fate of CD8 T cells specific for H60 as a model of CD8 T cell tolerance to a self antigen whose expression is restricted to the hematopoietic system. They find that clonal deletion of such CD8 T cells is incomplete, leading to the escape of low avidity T cells (as identified by lower tetramer staining). In h60- > h60+ BM chimeras (used here as a model of minor MHC antigen mismatched BM transplantation), they also describe incomplete negative selection. Escaping h60 specific CD8 T cells can acquire effector function and kill h60+ cells, both in a TCR transgenic system and in the polyclonal repertoire.

Overall, the MS unfortunately suffers from an overload with unnecessary and sometimes odd details, e.g. Figure 3 argues that deletion in Act H60 mice occurs at an early DN stage. This can only be an artefact of the TCR transgenic system, as DN cells do oftentimes prematurely express the transgenic TCR, whereas normal DN cells do not yet express the fully assembled ab TCR. Another example is Fig 4f-h: Here, BM reconstitution of b2m donors is performed and the fate of H60 specific cells is addressed. Of note, these should not even be positively selected in this constellation, so any interpretation of tolerogenic mechanisms is extremely problematic. In sum, it remains doubtful whether any of the findings in this MS can be generalized to other hematopoietic antigens.

Reviewer #3 (Remarks to the Author):

Ju and colleagues show that CD8 T cells specific for the hematopoietic cell-restricted antigen H60 can escape the negative selection in the thymus and drive eradication of hematopoietic and tumor cells harboring this antigen. Using different mouse models the authors investigate specifically the role of H60 in the negative selection process in the thymus and clarify at which step the selection occurs most likely. Difficult to perform mouse transplantation and chimera studies strengthened the work. Experiments were performed carefully with adequate controls.

By using the chimeric mouse model and BMT approach OT-I \rightarrow [Ova Tg \rightarrow B6], they could show that the negative selection in the thymus is applicable to other HRA-specific CD8 T cells under BMT conditions.

A methodological limitation is that the only tumor model they use is EL4 a T-lymphoblastic lymphoma. This is rarely the indication for allogeneic stem cell transplantation while myeloid

tumors such as acute myeloid leukemia are most frequent indications. Therefore the clinical relevance is somehow questionable. Also it remains unclear if in humans similar to the congenic mouse system hematopoietic specific polyclonal escapee CD8 T cells contribute to GVL effects.

Specific comments:

1. An explanation for the reduced avidity of the J15 cells after transplantation into Con-H60 mice is missing.
2. Why do the escapee CD8+ J15 cells do have a reduced avidity? How can the TCR in the Rag-1 -/-J15 \times Con-H60 be different from the TCR in the Rag-1 -/-J15 \times B6 mouse? There should be a single clonogenic population; so what is leading to the reduced avidity? TCR sequencing might be helpful
3. Did the authors evaluate if the TCR surface levels are influenced by staining for TCR β ?
4. To check at which stage the negative selection occurs, the authors checked for CD5 signaling. Would it be possible to check in addition by Tetramer-H60 staining at which stage the H60-specific T cells (with high avidity) in Con-H60 mice are eliminated
5. To characterize the J15 in B6 and Con-H60 recipients the authors analyzed IFN-gamma levels of H60-Tetramer+ CD8 T cells and cytotoxicity, which is different between escapee CD8 T cells of Con-H60 recipients and B6. In addition it would be interesting to know, if the proliferation rate is the same and how frequent regulatory cells are.
6. What is the reason for the increased infiltration of J15 cells into the intestine of Con-H60 mice?
7. Are also H60-Tet negative cells infiltrating?
8. Is the increased amplification of ex vivo expanded CD8+ thymocytes with H60+ feeder cells reflecting the lower in vivo frequencies or does the Con-H60 derived cells have a slower proliferation rate?
9. Besides EL4 other leukemia models need to be used to show that the findings are reproducible in other hematological malignancies and not restricted to one model.
10. The GVHD scoring was done according to Ref 49, however this is not a publication specifically on GVHD scoring. Other established scoring systems are recommended (e.g. Kaplan et al. J Immunol.)
11. How polyclonal are the H60-specific T cells? A recent study has shown that more polyclonal T cells are responsible for GVHD while rather oligoclonal T cells mediate GVL effects: van Bergen, C.A., van Luxemburg-Heijs, S.A., de Wreede, L.C., Eefting, M., von dem Borne PA, van Balen P, Heemskerk MH, Mulder A, Claas FH, Navarrete MA, Honders WM, Rutten CE, Veelken H, Jedema I, Halkes CJ, Griffioen M, Falkenburg JH. J Clin Invest 127, 2017

Minor:

Typo in Line 141 "medullar of the thymus" the "r" should be deleted.

Line 143: "mTEC-derived antigen-specific thymocytes" is puzzling. I would favor mTEC-selected

Line 302: patients suffering from what?

Point-by-Point Reply to Reviewers' Comments

Reviewer #1' comment.

The authors provide strong data that H60-specific T-cells escape thymic deletion, and that these cells can mediate GVL effects in the periphery. However, what is much less clear is the nature of the intrathymic cell that is expressing H60 to impose tolerance. The model proposed relies heavily on the notion that H60 is expressed exclusively by haemopoietic cells and not medullary thymic epithelium.

Given that many so-called tissue restricted antigens are now known to be expressed by mTEC, can the authors provide clear evidence that H60 is expressed only by haemopoietic cells, and be absent from mTEC? If not, then the data is compatible with the notion that mTEC may be mediating tolerance, rather than haemopoietic cells.

We are grateful for the reviewer's commendation of our work. We acknowledge the reviewer's point regarding experimental evidence for hematopoietic cell restricted distribution of H60 within the thymus. Therefore, we isolated the thymic dendritic cells (tDCs) and thymic medullary epithelial cells (mTECs) from Con-H60 mice by FACS sorting, and examined the presence of H60 mRNA in these populations by RT-PCR. H60 transcripts were detected in tDCs (hematopoietic cells), but not in mTECs, which supports our observations. In contrast, H60 transcripts were detected in both tDCs and mTECs from Act-H60 mice as expected based on the ubiquitous expression of H60 in these mice. No expression of H60 transcript in either population was detected in B6 mice, which served as the negative control. We provide these data as evidence of exclusive expression of H60 in hematopoietic APCs in the thymus of Con-H60 mice. These data are shown in new **Supplementary Fig. 5 b,c** and are described briefly in the Results section (Page 9; Line 199).

Reviewer #2 comments

The authors address the fate of CD8 T cells specific for H60 as a model of CD8 T cell tolerance to a self antigen whose expression is restricted to the hematopoietic system. They find that clonal deletion of such CD8 T cells is incomplete, leading to the escape of low avidity T cells (as identified by lower tetramer staining). In h60- > h60+ BM chimeras (used here as a model of minor MHC antigen mismatched BM transplantation), they also describe incomplete negative selection. Escaping h60 specific CD8 T cells can acquire effector function and kill h60+ cells, both in a TCR transgenic system and in the polyclonal repertoire. Overall, the MS unfortunately suffers from an overload with unnecessary and sometimes odd details, e.g. Figure 3 argues that deletion in Act H60 mice occurs at an early DN stage. This can only be an artefact of the TCR transgenic system, as DN cells do oftentimes prematurely express the transgenic TCR, whereas normal DN cells do not yet express the fully assembled ab TCR. Another example is Fig 4f-h: Here, BM reconstitution of b2m donors is performed and the fate of H60 specific cells is addressed. Of note, these should not even be positively selected in this constellation, so any interpretation of tolerogenic mechanisms is extremely problematic. In sum, it remains doubtful whether any of the findings in this MS can be generalized to other hematopoietic antigens.

We appreciate the detailed comments on our manuscript. Point-by-point responses to the comments of the reviewer are provided below.

- “Overall, the MS unfortunately suffers from an overload with unnecessary and sometimes odd details, e.g. Figure 3 argues that deletion in Act H60 mice occurs at an early DN stage. This can only be an artefact of the TCR transgenic system, as DN cells do oftentimes prematurely express the transgenic TCR, whereas normal DN cells do not yet express the fully assembled ab TCR.”
- ⇒ We agree that the negative selection at the DN stage of J15 TCR-transgenic T cell development does not represent the negative selection of usual polyclonal T cells. However, many TCR-transgenic mouse strains with this DN selection phenotype (e.g. HY-TCR transgenic mice) contributed to our current understanding of the role of APCs in thymic deletion.
- ⇒ The main focus of our manuscript is whether hematopoietic cell-restricted antigen (HRA)-specific T cells would be completely deleted in the thymus or not, rather than identifying the specific stage at which they are deleted. In other words, J15 T cells represent a model T cell that expresses a TCR specific for an HRA and the focus will be on identifying APC populations that can or cannot delete this HRA-specific T cells. Figure 3 shows preliminary data that facilitated identification of APCs that can delete the HRA-specific T cells. Figure 3 provides basic information regarding J15 system, such

as CD5 expression and DN4/DN ratios, that is necessary for interpreting the experimental data presented in Figure 4 and identifying the APCs responsible for HRA-specific T cells.

⇒ Moreover, similarly incomplete deletion was observed in an alternative model of HRA (new Supplementary Fig. 3), in which OVA is expressed by hematopoietic thymic APCs and OT-1 TCR-Tg T cells are selected in the thymus. Finally, the generation of deletion escapee T cells specific for H60 HRA is also observed in a polyclonal setting, which represents physiological thymic selection environment (Fig. 7). Thus, incomplete deletion of HRA-specific T cells in the thymus is not due to “abnormal” deletion kinetics of the J15 system. The J15 system in Figures 3 and 4 provides benefits of magnifying and analyzing APC-mediated deletion of HRA-specific T cells, which cannot be done in a polyclonal setting with too few cells for analysis.

■ “Another example is Fig 4f-h: Here, BM reconstitution of b2m donors is performed and the fate of H60 specific cells is addressed. Of note, these should not even be positively selected in this constellation, so any interpretation of tolerogenic mechanisms is extremely problematic.”

⇒ The point of this comment is unclear to us.

First, Fig. 4f-h shows the results of experiments using the BM from the wild-type J15 donors, not b2m-deficient donors. If the reviewer is referring to Fig 4f, the b2m-deficient J15 → b2m-deficient B6 BMT set is the negative control for immunostaining of b2m-positive cells in the thymus and does not represent actual biological data.

⇒ If the reviewer is referring mistakenly to Fig 4a-d instead of Fig 4f-h, b2m-deficient donor BM-derived T cells can undergo positive selection in the b2m-sufficient host, using b2m-positive host hematopoietic and epithelial APCs, which T cell immunologists would likely agree with. If b2m-deficient donor BM is transplanted to a b2m-deficient host, then the positive selection does not occur, as the reviewer states. We have added data relevant to this situation (i. e., lack of positive selection) as a control ($\beta 2m^{-/-}$ J15 → $\beta 2m^{-/-}$ B6) to the revised **Fig. 4a, b** for clarification.

- “In sum, it remains doubtful whether any of the findings in this MS can be generalized to other hematopoietic antigens.”
- ⇒ As mentioned above, we included the Ova Tg → B6 BMT set to mimic hematopoietic cell-restricted Ova expression in BM recipients (new Supplementary Fig. 3), because no appropriate natural hematopoietic cell-restricted antigen and specific TCR Tg system other than H60 is known. In the OT-1 → [Ova Tg→B6] BMT set, OT-1 thymocytes are also incompletely deleted. Thus, incomplete deletion of HRA-specific thymocytes is not unique to H60, but can be generalized to at least two different HRAs. This interpretation is supported by reviewer #3 as follows: “By using the chimeric mouse model and BMT approach OT-I [Ova Tg B6], they could show that the negative selection in the thymus is applicable to other HRA-specific CD8 T cells under BMT conditions.”

Reviewer #3 comments

Ju and colleagues show that CD8 T cells specific for the hematopoietic cell-restricted antigen H60 can escape the negative selection in the thymus and drive eradication of hematopoietic and tumor cells harboring this antigen. Using different mouse models the authors investigate specifically the role of H60 in the negative selection process in the thymus and clarify at which step the selection occurs most likely. Difficult to perform mouse transplantation and chimera studies strengthened the work. Experiments were performed carefully with adequate controls.

By using the chimeric mouse model and BMT approach OT-I [Ova Tg B6], they could show that the negative selection in the thymus is applicable to other HRA-specific CD8 T cells under BMT conditions.

A methodological limitation is that the only tumor model they use is EL4 a T-lymphoblastic lymphoma. This is rarely the indication for allogeneic stem cell transplantation while myeloid tumors such as acute myeloid leukemia are most frequent indications. Therefore the clinical relevance is somehow questionable. Also it remains unclear if in humans similar to the congenic mouse system hematopoietic specific polyclonal escapee CD8 T cells contribute to GVL effects.

Specific comments:

1. An explanation for the reduced avidity of the J15 cells after transplantation into Con-H60 mice is missing.
2. Why do the escapee CD8+ J15 cells do have a reduced avidity? How can the TCR in the Rag-1 -/-J15 Con-H60 be different from the TCR in the Rag-1 -/-J15 B6 mouse? There should be a single clonal population; so what is leading to the reduced avidity? TCR sequencing might be helpful
3. Did the authors evaluate if the TCR surface levels are influenced by staining for TCRB?
4. To check at which stage the negative selection occurs, the authors checked for CD5 signaling. Would it be possible to check in addition by Tetramer-H60 staining at which stage the H60-specific T cells (with high avidity) in Con-H60 mice are eliminated
5. To characterize the J15 in B6 and Con-H60 recipients the authors analyzed IFN-gamma levels of H60-Tetramer+ CD8 T cells and cytotoxicity, which is different between escapee CD8 T cells of Con-H60 recipients and B6. In addition it would be interesting to know, if the proliferation rate is the same and how frequent regulatory cells are.
6. What is the reason for the increased infiltration of J15 cells into the intestine of Con-H60 mice?
7. Are also H60-Tet negative cells infiltrating?
8. Is the increased amplification of ex vivo expanded CD8+ thymocytes with H60+ feeder cells reflecting the lower in vivo frequencies or does the Con-H60 derived cells have a slower proliferation rate?
9. Besides EL4 other leukemia models need to be used to show that the findings are reproducible in other hematological malignancies and not restricted to one model.
10. The GVHD scoring was done according to Ref 49, however this is not a publication specifically on GV

HD scoring. Other established scoring systems are recommended (e.g. Kaplan et al. J Immunol.)

11. How polyclonal are the H60-specific T cells? A recent study has shown that more polyclonal T cells are responsible for GVHD while rather oligoclonal T cells mediate GVL effects: van Bergen, C.A., van Luxemburg-Heijs, S.A., de Wreede, L.C., Eefting, M., von dem Borne PA, van Balen P, Heemskerk MH, Mulder A, Claas FH, Navarrete MA, Honders WM, Rutten CE, Veelken H, Jedema I, Halkes CJ, Griffioen M, Falkenburg JH. J Clin Invest 127, 2017

Minor:

Typo in Line 141 "medullar of the thymus" the "r" should be deleted.

Line 143: "mTEC-derived antigen-specific thymocytes" is puzzling. I would favor mTEC-selected

Line 302: patients suffering from what?

We appreciate the reviewer's acknowledgement of our work and the comments which we feel helped us to improve our manuscript. Responses to the reviewer's comments are provided below.

- 1. An explanation for the reduced avidity of the J15 cells after transplantation into Con-H60 mice is missing.
 - 2. Why do the escapee CD8⁺ J15 cells do have a reduced avidity? How can the TCR in the Rag-1^{-/-}-J15 Con-H60 be different from the TCR in the Rag-1^{-/-}-J15 B6 mouse? There should be a single clonogenic population: so what is leading to the reduced avidity? TCR sequencing might be helpful.
- ⇒ We agree that we need to state the reason for the reduced avidity; i. e., the low intensity tetramer-staining of the escapee J15 cells. There are two possible explanations for this phenomenon. First, in [J15 X Con-H60] F1 mice and J15 → Con-H60 BMT recipients, the J15 TCR β-chain can pair with endogenous α-chains rather than the J15 α-chain due to incomplete allelic exclusion. Although many of these T cells with endogenous TCR α-chain expression were tetramer-negative (Fig 1a and 2b), some were reactive to H60 with low avidity and were positively selected. Indeed, substantial percentages of tetramer-positive CD8 T cells generated in Con-H60 hosts harbored endogenous α-chains when stained with Vα subtype-specific antibodies (approximately 18% in total for Vα2, Vα3.2 and Vα8.3-positive cells in H60-tetramer⁺ T cells; new **Supplementary Fig 2a, c**). Thus, endogenous α-chain-harboring J15 cells likely constitute a proportion of the low avidity escapee J15 cells.
- ⇒ However, as stated by the reviewer in comment #2, in Rag-1^{-/-} J15 → Con-H60 BMT recipients, in which all endogenous α-chain-harboring T cells were eliminated due to Rag-1 deficiency and thus most CD8 T cells were tetramer-positive (Fig 2c), tetramer-positive cells retained the reduced tetramer-staining intensity. The monoclonality of Rag-1^{-/-} J15 Con-H60 and Rag-1^{-/-} J15 B6 cells was confirmed by the absence of endogenous α-chain-expressing T cells (new **Supplementary Fig 2c**) and sole detection of transgenic Vα-chain (Vα10.3) mRNA, but not other Vα-chain mRNAs (data not shown). (Antibody to Vα10.3 is not commercially available, which precludes staining of Vα10.3-harboring cells.) Thus, mechanisms other than endogenous α-chain J15 TCR also seem to be in play. One possible explanation is that J15 escapee T cells showed reduced expression levels of TCR and CD8 on their surface in all cases, including [J15 X Con-H60] F1 mice, and J15 → Con-H60 and Rag-1^{-/-} J15 → Con-H60 BMT recipients, compared to those on their B6 counterparts (new **Supplementary Fig 2b, d**). TCR and CD8 are critical determinants for binding of peptide-MHC tetramers. Downregulation of TCR and CD8 on TCR Tg T cells after recognition of cognate antigen in the thymus and periphery has been reported (Cell 65: 292, 1991;

Science 251: 1225, 1991; Immunity 3: 407, 1995). Therefore, we propose that some of the intact J15 cells with transgenic α and β chains down-regulate their surface TCR and CD8 expression upon activation during thymic selection and survive to become low-avidity T cells. This low-avidity seems to be maintained by continuous antigenic stimulation both in the thymus and periphery, because upregulation of CD69 and/or CD44 was observed in both thymic and splenic tetramer-positive J15 T cells in Con-H60 hosts, but not in B6 hosts (new **Supplementary Fig 2b,d**).

These results and interpretations are presented in the new **Supplementary Fig. 2**, and described in the text on pages 6 and 7.

■ 3. Did the authors evaluate if the TCR surface levels are influenced by staining for TCR β ?

⇒ We understand the concern over whether the reduced tetramer-staining intensity is an artifact generated by overlap between the binding epitopes of anti-TCR V β 8.3 antibody and H60-tetramers. To assess this, we compared the fluorescence intensities of H60-tetramer-staining and general anti-TCR β antibody-staining in the presence and absence of anti-V β 8.3 antibody. The results (provided below) confirmed that TCR and tetramer-staining intensities were not influenced by the presence of the anti-V β 8.3 antibody.

- 4. To check at which stage the negative selection occurs, the authors checked for CD5 signaling. Would it be possible to check in addition by tetramer-H60 staining at which stage the H60-specific T cells (with high avidity) in Con-H60 mice are eliminated?

⇒ This comment assisted determination of the stage at which J15 thymocytes are deleted in Con-H60 and Act-H60 recipients. Because peptide-MHC-tetramers do not bind to the TCR in the absence of CD8 expression on T cells, it is not possible to use tetramer-staining to detect DN-stage thymocytes. Therefore, we used an anti-active Caspase 3 antibody, to identify DN cells undergoing deletional apoptosis. Significant proportions of DN4 and DP cells from Con-H60 recipients, and DN2 cells from Act-H60 recipients were positive for active Caspase 3 expression. Because DN4-DP cells and DN2 cells were the stages at which CD5 was upregulated in the thymi of Con-H60 and Act-H60 recipients, respectively, this finding demonstrates that J15 thymocytes are subjected to deletion at the CD5 upregulation stage in both negative selection conditions. The absence of active Caspase 3 staining at the DP stage in the thymus of B6 mice indicates that J15 cells are positively selected at this stage. These results have been added to the revised manuscript in the form of the new **Fig. 3c** and are described in the text (Page 8; Line 165).

- 5. To characterize the J15 in B6 and Con-H60 recipients, the authors analyzed IFN-gamma levels of H60-tetramer⁺ CD8 T cells and cytotoxicity, which is different between escapee CD8 T cells of Con-H60 recipients and B6. In addition, it would be interesting to know, if the proliferation rate is the same and how frequent regulatory cells are.

⇒ This comment enhanced our understanding of the characteristics of thymic deletion escapees. To address the proliferation rates, we performed BrdU-incorporation assays using the H60-tetramer⁺ splenic CD8 T cells from Con-H60 or B6 recipients of J15 BM. BrdU was treated at 24 h after initiation of mixed leukocyte culture (MLC) of splenic CD8 T cells with H60⁺ feeder cells, in the presence of a low concentration of IL-2 (10 U/ml). Most H60-tetramer⁺ CD8 T cells from Con-H60 recipients of J15 BM incorporated BrdU up to 72 h, although they showed delayed kinetics of BrdU-incorporation compared

to that of H60-tetramer⁺ CD8 T cells from B6 recipients, which is consistent with their lower IFN- γ production compared to that of the latter. This indicates that escapee CD8 T cells, despite their relatively low avidity for H60, have the potential to proliferate in response to the cognate antigen as well as the capacity to perform effector functions. These data are shown in new **supplementary Fig. 7a** and mentioned in the Result section (Page 10; Line 218).

⇒ Regarding the regulatory cells, we did not find any difference in the proportion of Foxp3⁺ populations in splenic and intestinal T cells from Con-H60 and B6 recipients of J15 BM. These data are presented as **supplementary Fig. 7c** and are described in the text (Page 11; Line 236)

■ 6. What is the reason for the increased infiltration of J15 cells into the intestine of Con-H60 mice ?

⇒ Although the reason is unclear, we believe that the intestinal inflammation induced by total body irradiation at the time of BMT and/or local environmental factors contributed to this T cell recruitment. The following experimental observations support this interpretation.

⇒ Newly generated T cells infiltrated into the peripheral organs, including the intestine, of all three BM recipients (Fig. 5c). In the Con-H60 recipients, H60-transcripts were detected in the lysates of leukocytes in the intestine at 4 weeks after J15 BMT, a time point when *de novo* generated J15 T cells began to emigrate to the periphery. Thus, the H60-positive leukocytes in the early inflamed intestine (probably due to total body irradiation) present H60 to newly generated J15 T cells (escapees), inducing T cell-mediated inflammation. The T cell-mediated aggravation of tissue inflammation results in increased recruitment of T cells to the intestine. In contrast, despite the presence of H60-positive leukocytes in the intestine, Act-H60 recipients did not show intestinal inflammation due to the complete thymic negative selection of J15 T cells. B6-recipients did not develop T-cell mediated intestinal inflammation either, because they lack H60 antigen. The RT-PCR data showing the presence of H60-positive leukocytes in the intestine are shown in new **Supplementary Fig. 8d** and described briefly in the text (Page 12; Line 252).

⇒ The preferential infiltration of J15 cells to the gut over other organs, e. g., the liver, in Con-H60 recipients

may be explained by preferential expression of gut-homing receptors on tetramer-positive T cells in the periphery. Splenic H60-tetramer-positive CD8 T cells in Con-H60 recipients expressed the gut-homing receptor $\alpha 4\beta 7$ at significantly higher frequencies compared with splenic H60-tetramer-negative CD8 T cells in Con-H60 recipients, or splenic H60-tetramer-positive/or -negative CD8 T cells in B6 recipients. However, the frequency of liver-homing receptor CXCR6⁺ cells was not significantly different among splenic H60-tetramer-positive and -negative populations in Con-H60 or B6 recipients. This is shown in new **Supplementary Fig. 8e** and described in the text (Page 12; Line 253). Thus, mild intestinal inflammation led to the recruitment of H60-positive leukocytes, and this antigenic localization and tissue-specific chemotactic signaling contributed to J15 T cell-mediated inflammation in the intestine of Con-H60 hosts.

⇒ The factors that activate expression of gut-homing receptors on J15 cells are unclear. Preferential recruitment to the intestine over the liver indicates that gut-specific inflammatory factors, such as the microbiota, might have contributed to this phenomenon. This was described in the Discussion section of the original manuscript as follows: “...inflammatory signals generated due to lethal irradiation and the intestinal tissue environment play roles in enhancing the infiltration and effector differentiation of the H60-cognate escapee CD8 T cells specifically in the intestines” (Page 17; Line 375).

■ 7. Are also H60-Tet negative cells infiltrating ?

⇒ Yes, they are; however, tetramer-negative cells are a minor population among T cells infiltrating the intestines. This information is shown in new **Supplementary Fig. 7b** and mentioned briefly in the text (Page 10; Line 225).

■ 8. In the increased amplification of ex vivo expanded CD8⁺ thymocytes with H60⁺ feeder cells reflecting the lower in vivo frequencies or does the Con-H60-derived cells have a slow proliferation rate ?

⇒ CD8⁺ thymocytes and splenocytes from Con-H60 mice showed lower rates of BrdU-incorporation during MLC, compared with their B6 counterparts. Because the *ex vivo* frequencies of tetramer⁺ cells differed

significantly between Con-H60 and B6 mice, we believe that the lower frequency values obtained after *in vitro* amplification of the H60-tetramer⁺ CD8 T cells from Con-H60 mice are due to both lower initial frequencies *in vivo* and slower proliferation *in vitro* compared to those from B6 mice. The results of BrdU incorporation assays are shown in new **Supplementary Fig. 9c** and described in the text (Page 13; Line 283).

- 9. Besides EL4, other leukemia models need to be used to show that the findings are reproducible in other hematological malignancies and not restricted to one model.

⇒ We agree that another tumor model would enable generalization of our findings regarding the GVL effects exerted by the hematopoietic antigen-specific deletion escapees. Therefore, we used C1498 cells, B6 myeloid leukemia cells, to test the GVL effect. All findings observed with EL4 cells were recapitulated with C1498 leukemic cells. Con-H60 or B6 recipients of H60⁺C1498 cells survived for a long time, and H60-tetramer⁺ cells were detected in the PBLs, while Act-H60 recipients injected with H60⁺C1498 cells died within 24 days post-tumor cell injection. All three types of recipients injected with H60⁺C1498 cells died prior to 30 days post-tumor cell injection. These results are described in the text (Page 14; Line 314) and shown in new **Fig. 8f, g**.

- 10. The GVHD scoring was done according to Ref 49. However, this is not a publication specifically on GVHD scoring. Other established scoring systems are recommended.

⇒ We re-evaluated GVHD scores as suggested by the reviewer. Because pathology was obvious in the intestine tissue sections of Con-H60 recipients that exhibited bodyweight loss, the new scoring system did not alter the significant differences in the overall pathological scores of Con-H60 recipients and the two other recipients. We have added this article to the References section.

- 11. How polyclonal are the H60-specific T cells? A recent study has shown that more polyclonal T cells

are responsible for GVHD while rather oligoclonal T cells mediate GVL effects.

- ⇒ The results from the human study mentioned by the reviewer are likely not relevant here, because of the differences in conditions between the two studies. The human study investigated the clonality of T cells from patients with GVL or GVHD under which circumstances dynamic interplay exists among T cells with different antigen specificities, leading to the immunodominance phenomenon. Therefore, the conclusion from the human study cannot be applied to our work in which peripheral activity of single antigen-specific T cells was tested. Our assumption was that full diversity of T cells would be present within the single antigen-specific T cell pool, rather than competition with T cells of different antigen specificities, based on the following previous our studies that we conducted on H60.
- ⇒ We reported a high frequency of H60-specific T cells in a B6 naïve CD8 T cell pool (Choi et al, Immunity 17:593, 2002) and use of various TCRs by CD8 T cells responding to H60 (Transplantation 87:1609, 2009; Molecules and Cells 33:393, 2012). Therefore, H60-specific T cells generated in the B6 environment are polyclonal.
- ⇒ Nonetheless, the reviewer's point on the possible change in the polyclonality of H60-specific T cells under the incomplete thymic deletion system in this study seems valid. Therefore, we analyzed TCR β chain usage of H60-tetramer-binding CD8 T cells isolated from the spleens of B6 or Con-H60 recipients of CD45.1⁺B6 BM, using a next generation sequencing (NGS) technique. The sequencing data revealed that H60-tetramer⁺ T cells from Con-H60 recipients use diverse TCR β chains, similar to those from B6 recipients, when analyzed using the Shannon entropy and Simpson indexes. The data are provided below.

- ⇒ Thus, polyclonality of H60-specific T cells is maintained in the incomplete thymic deletion condition in this study. Although we do not think that these data are critical for understanding the main theme of our

study, if the reviewer or the editor think that they need to be included in the manuscript, we are happy to provide them as a supplementary information.

Minor

- Typo in line 141 “medullar of the thymus” the “r” should be deleted

⇒ This error has been corrected in the revised manuscript.

- Line 143: mTEC-derived antigen-specific thymotes is puzzling. I would favor mTEC-selected

⇒ We are grateful for this suggestion; the manuscript has been revised accordingly.

- Line 302: patients suffering from what

⇒ To clarify, we have revised this text to “ leukemia patients who received allo-BMT”.

REVIEWERS' COMMENTS:

Reviewer #1 (Remarks to the Author):

The authors provide convincing data on the expression pattern of self antigen in thymic DC and mTEC

Reviewer #3 (Remarks to the Author):

The authors have fully addressed my technical concerns.